# A screen of drug-like molecules identifies chemically diverse electron transport chain inhibitors in apicomplexan parasites

Jenni A. Hayward[ID][¤a◉], F. Victor Makota[ID][◉], Daniela Cihalova, Rachel A. Leonard, Esther Rajendran[ID], Soraya M. Zwahlen[ID][¤b], Laura Shuttleworth[¤c], Ursula Wiedemann, Christina Spry[ID], Kevin J. Saliba[ID]*, Alexander G. Maier[ID]*, Giel G. van Dooren[ID]*

Research School of Biology, Australian National University, Canberra, Australia

◉ These authors contributed equally to this work.
¤a Current address: Monash Biomedicine Discovery Institute, Department of Biochemistry and Molecular Biology, Monash University, Clayton, Australia
¤b Current address: Developmental Biology Unit, European Molecular Biology Laboratory, Heidelberg, Germany
¤c Current address: The Francis Crick Institute, London, United Kingdom
* kevin.saliba@anu.edu.au (KJS); alex.maier@anu.edu.au (AGM); giel.vandooren@anu.edu.au (GGvD)

**Data Availability Statement:** All relevant data are within the manuscript and its Supporting Information files.

## Abstract

Apicomplexans are widespread parasites of humans and other animals, and include the causative agents of malaria (*Plasmodium* species) and toxoplasmosis (*Toxoplasma gondii*). Existing anti-apicomplexan therapies are beset with issues around drug resistance and toxicity, and new treatment options are needed. The mitochondrial electron transport chain (ETC) is one of the few processes that has been validated as a drug target in apicomplexans. To identify new inhibitors of the apicomplexan ETC, we developed a Seahorse XFe96 flux analyzer approach to screen the 400 compounds contained within the Medicines for Malaria Venture 'Pathogen Box' for ETC inhibition. We identified six chemically diverse, on-target inhibitors of the ETC in *T. gondii*, at least four of which also target the ETC of *Plasmodium falciparum*. Two of the identified compounds (MMV024937 and MMV688853) represent novel ETC inhibitor chemotypes. MMV688853 belongs to a compound class, the aminopyrazole carboxamides, that were shown previously to target a kinase with a key role in parasite invasion of host cells. Our data therefore reveal that MMV688853 has dual targets in apicomplexans. We further developed our approach to pinpoint the molecular targets of these inhibitors, demonstrating that all target Complex III of the ETC, with MMV688853 targeting the ubiquinone reduction ($Q_i$) site of the complex. Most of the compounds we identified remain effective inhibitors of parasites that are resistant to Complex III inhibitors that are in clinical use or development, indicating that they could be used in treating drug resistant parasites. In sum, we have developed a versatile, scalable approach to screen for compounds that target the ETC in apicomplexan parasites, and used this to identify and characterize novel inhibitors.

**Funding:** This work was supported by a Research School of Biology innovation grant to ER, DC, AGM and GGvD, a National Health and Medical Research Council Ideas Grant (GNT1182369) to GGvD, AGM and KJS, an Australian Research Council Discovery project (DP180103212) to AGM, and an Australian Government Research Training Program Scholarship to JAH. The funders had no role in study design, data collection and analysis, decision to publish, or preparation of the manuscript. The authors declare that they have no conflicts of interest.

**Competing interests:** The authors have declared that no competing interests exist.

## Author summary

Apicomplexan parasites impart major health and economic burdens on human societies. Treatment options against these parasites, which include the causative agents of toxoplasmosis (*Toxoplasma gondii*) and malaria (*Plasmodium* spp.), are limited. The apicomplexan mitochondrial electron transport chain is critical for parasite proliferation and pathogenesis, and is a validated drug target. In this study, we develop a powerful suite of approaches for screening compound libraries to identify and characterize electron transport chain inhibitors in these parasites that are potent, chemically diverse, and active against drug resistant strains of the parasites. These approaches enable us to distinguish between on-target inhibitors and those that cause non-specific parasite death, and allow us to pin-point the molecular target of inhibitors. We employ these approaches to identify an inhibitor with dual molecular targets. The novel compounds we identify represent new pathways towards much needed treatments for the diseases caused by apicomplexans.

## Introduction

Apicomplexan parasites cause numerous diseases in humans and livestock worldwide. Up to a third of the global human population is chronically infected with *Toxoplasma gondii*, which can cause the disease toxoplasmosis in immunocompromised individuals and developing fetuses [1]. *Plasmodium* parasites cause the disease malaria, which killed over 600,000 people and infected ~240 million in 2020 [2]. Despite the recent approval of the first malaria vaccine for children by the World Health Organization, there is currently no effective vaccine against malaria for adults or against toxoplasmosis in humans. There is therefore a heavy reliance on drugs to treat both diseases. Current treatment options are limited and have questionable efficacy and safety. For instance, while frontline therapeutics such as pyrimethamine and sulfadiazine are able to kill the disease-causing tachyzoite stage of *T. gondii*, they elicit adverse effects in many patients and fail to eradicate the long-lived bradyzoite cyst stage that causes chronic infection [3]. Emerging resistance to frontline therapeutics, such as artemisinin, is a particular problem for treating the potentially life-threatening severe malaria caused by *Plasmodium falciparum* [4]. New treatments for toxoplasmosis and malaria are therefore much needed.

The mitochondrion is important for apicomplexan parasite survival and is a target of many anti-parasitic compounds [5]. Like in other eukaryotes, the inner membrane of the parasite mitochondrion houses an electron transport chain (ETC), which is composed of a series of protein complexes that contribute to energy generation and pyrimidine biosynthesis [6]. Electrons derived from parasite metabolism are fed into the ETC via the action of several dehydrogenases–including succinate dehydrogenase (SDH), malate-quinone oxidoreductase (MQO), glycerol 3-phosphate dehydrogenase (G3PDH), dihydroorotate dehydrogenase (DHODH), and type II NADH dehydrogenases (NDH2)–which all reduce the hydrophobic inner membrane electron transporting molecule coenzyme Q (CoQ). CoQ interacts with ETC Complex III (also known as the coenzyme Q:cytochrome *c* oxidoreductase or $bc_1$ complex) at the so-called CoQ oxidation ($Q_o$) and CoQ reduction ($Q_i$) sites, where electrons are donated to or accepted from Complex III, respectively, in a process termed the Q cycle [7]. This process also contributes to the generation of a proton motive force across the inner mitochondrial membrane by facilitating the net translocation of protons from the matrix into the intermembrane space. Complex III passes electrons to the soluble intermembrane space protein cytochrome *c* (CytC). CytC shuttles the electrons to ETC Complex IV (cytochrome *c* oxidase), which donates them to the terminal electron acceptor, molecular oxygen ($O_2$). Complex IV also contributes

to the proton motive force by translocating protons across the inner mitochondrial membrane. The net reaction of the ETC is thus the oxidation of cellular substrates and reduction of $O_2$, coupled to the translocation of protons from the matrix into the intermembrane space to generate a proton gradient across the inner membrane. This proton gradient can be utilized by an F-type ATPase (Complex V) to generate ATP and for important mitochondrial processes such as protein import [8]. In the erythrocytic stages of the *P. falciparum* lifecycle, the ETC functions primarily as an electron sink for the DHODH reaction in the *de novo* pyrimidine biosynthesis pathway rather than for ATP synthesis [9].

ETC Complex III is the target of many anti-parasitic agents, including the clinically used therapeutic atovaquone and the pre-clinical 'endochin-like quinolone' (ELQ) compounds [10–12]. Many Complex III-targeting compounds are CoQ analogs that bind to the $Q_o$ and/or $Q_i$ sites of Complex III [13]. The ability of these compounds to selectively target parasite Complex III lies in differences in the CoQ binding site residues of the complex between parasites and the mammalian hosts they infect, specifically in the cytochrome *b* protein [14–16]. For instance, the $Q_o$ site inhibitor atovaquone has an $EC_{50}$ value in the nanomolar range against Complex III activity in *T. gondii* and *P. falciparum*, but inhibits the mammalian complex 13- to 230-fold less effectively [11,17,18]. Although it is a potent and selective inhibitor of Complex III in apicomplexans, resistance to atovaquone can readily emerge as the result of mutations in the cytochrome *b* protein [19,20], limiting its use in treating the diseases caused by these parasites. Identifying Complex III inhibitors that remain effective against atovaquone-resistant parasites is therefore desirable.

Strategies to identify new anti-parasitic compounds often use high throughput screening of small molecule libraries to identify inhibitors of parasite proliferation [21–24]. Adapting such high throughput screens to more specific assays offers a route to identifying inhibitors that target particular processes in the parasite. For example, researchers have exploited the observation that the *P. falciparum* ETC becomes dispensable when a cytosolic, CoQ-independent form of DHODH from yeast (yDHODH) is introduced into the parasite [9], to develop an indirect target-based screening approach [25]. This study identified compounds that have reduced potency against yDHODH-expressing parasites compared to WT *P. falciparum*, suggesting that they may be on-target inhibitors of the ETC of these parasites [25]. Parasite ETC inhibitors have been identified through screening of a compound library using a fluorescence-based Oxygen Biosensor System to directly measure $O_2$ consumption in erythrocytes infected with *Plasmodium yoelii* [26]. Although this approach is a powerful means of identifying candidate ETC inhibitors, shortcomings of this assay include that it has limited ability to distinguish between on-target ETC inhibitors and off-target compounds that cause parasite death (and therefore lead indirectly to decreased $O_2$ consumption) [26], and secondary assays are required to locate the target of identified inhibitors from these screens. An assay in which $O_2$ consumption and parasite viability could simultaneously be assessed in real-time would enable on- and off-target compounds to be differentiated more rigorously, and screening assays that pinpoint the molecular target(s) of candidate ETC inhibitors would provide a valuable means of identifying novel drug targets in the ETC.

We have recently established some versatile approaches to probe ETC function in apicomplexan parasites using a Seahorse XFe96 flux analyzer. These approaches enable us to simultaneously determine the parasite mitochondrial $O_2$ consumption rate (OCR), a measure of ETC activity, and the parasite extracellular acidification rate (ECAR), a proxy for parasite metabolic activity and viability, in real-time [27–29]. We have further adapted these assays to enable us to diagnose where in the ETC specific defects arise [28,29]. In this manuscript, we use these suite of Seahorse XFe96 flux analyzer assays to screen the Medicines for Malaria Venture (MMV) 'Pathogen Box' small molecule library for inhibitors of the *T. gondii* parasite ETC. We

identified seven compounds that inhibited *T. gondii* OCR, six of which were on-target ETC inhibitors, and a seventh that simultaneously inhibited ECAR, causing rapid parasite death in an off-target manner. Among these compounds were two chemically novel ETC inhibitors, one of which (MMV688853) was previously characterized as an inhibitor of the parasite calcium dependent protein kinase 1 (CDPK1) protein, and which our data therefore indicate has dual targets. We provide evidence that most of the identified inhibitors are also on-target inhibitors of the *P. falciparum* ETC, illustrating that these compounds have broad utility in targeting this important phylum of parasites. We utilized the Seahorse XFe96 flux analyzer assays to identify the targets of these inhibitors, and determined that most target ETC Complex III in these parasites. We also demonstrate that *T. gondii* and *P. falciparum* strains resistant to established ETC inhibitors show limited cross-resistance, and in some instances increased sensitivity, to some of the identified Complex III inhibitors. Taken together, our work establishes a scalable pipeline to both identify and characterize the targets of inhibitors of the ETC in apicomplexan parasites, providing new avenues towards much-needed treatments against these parasites.

## Results

### Screening the MMV 'Pathogen Box' identifies 7 inhibitors of $O_2$ consumption in *T. gondii*

Apicomplexan parasites require $O_2$ for one key purpose–to act as the terminal electron acceptor in the mitochondrial ETC. In previous studies, we utilized a Seahorse XFe96 extracellular flux analyzer assay to measure the mitochondrial $O_2$ consumption rate (OCR) in extracellular tachyzoites [27–29]. These assays enable the injection of compounds into wells of a 96-well plate prior to measuring parasite OCR, and we demonstrated that injection of the Complex III inhibitor atovaquone rapidly inhibits OCR [27,29]. We reasoned that this approach could be used to screen large compound libraries to identify new inhibitors of the parasite ETC. To investigate this, we screened the MMV 'Pathogen Box' compound library (a library of 'diverse, drug-like molecules active against neglected diseases') for inhibitors of parasite mitochondrial OCR in the disease-causing tachyzoite stage of *T. gondii* parasites. Of the 400 compounds tested, seven were found to inhibit OCR by more than 30% at 1 μM (Fig 1; S1 Table).

Chemically diverse compound scaffolds were represented among the identified hits (Fig 1), including the known apicomplexan parasite ETC inhibitors MMV689480 (buparvaquone) and the endochin-like quinolone (ELQ) family compound MMV671636. The anti-fungal agents MMV688754 and MMV021057 (trifloxystrobin and azoxystrobin, respectively) were also identified; these compounds bind to the $Q_o$ site of Complex III in fungi [30] and have been shown previously to inhibit *P. falciparum* proliferation [31], likely via binding to the $Q_o$ site of Complex III [32]. Other compounds identified in our screen have not yet been shown to be ETC inhibitors and included MMV688853 (an aminopyrazole carboxamide compound previously identified as an inhibitor of *T. gondii* calcium-dependent protein kinase 1 (*Tg*CDPK1); [33,34]), MMV024397 (which has been shown to inhibit proliferation of *P. falciparum*; [35]), and MMV688978 (auranofin). Auranofin is a gold-containing compound used clinically for the treatment of rheumatoid arthritis [36], which also inhibits the proliferation of many parasites including *T. gondii* [37] and *P. falciparum* [38].

### Identified compounds inhibit proliferation and $O_2$ consumption in both *T. gondii* and *P. falciparum*

We next tested whether the identified compounds could inhibit proliferation of *T. gondii* parasites. We measured the proliferation of RH strain *T. gondii* tachyzoites expressing a tandem

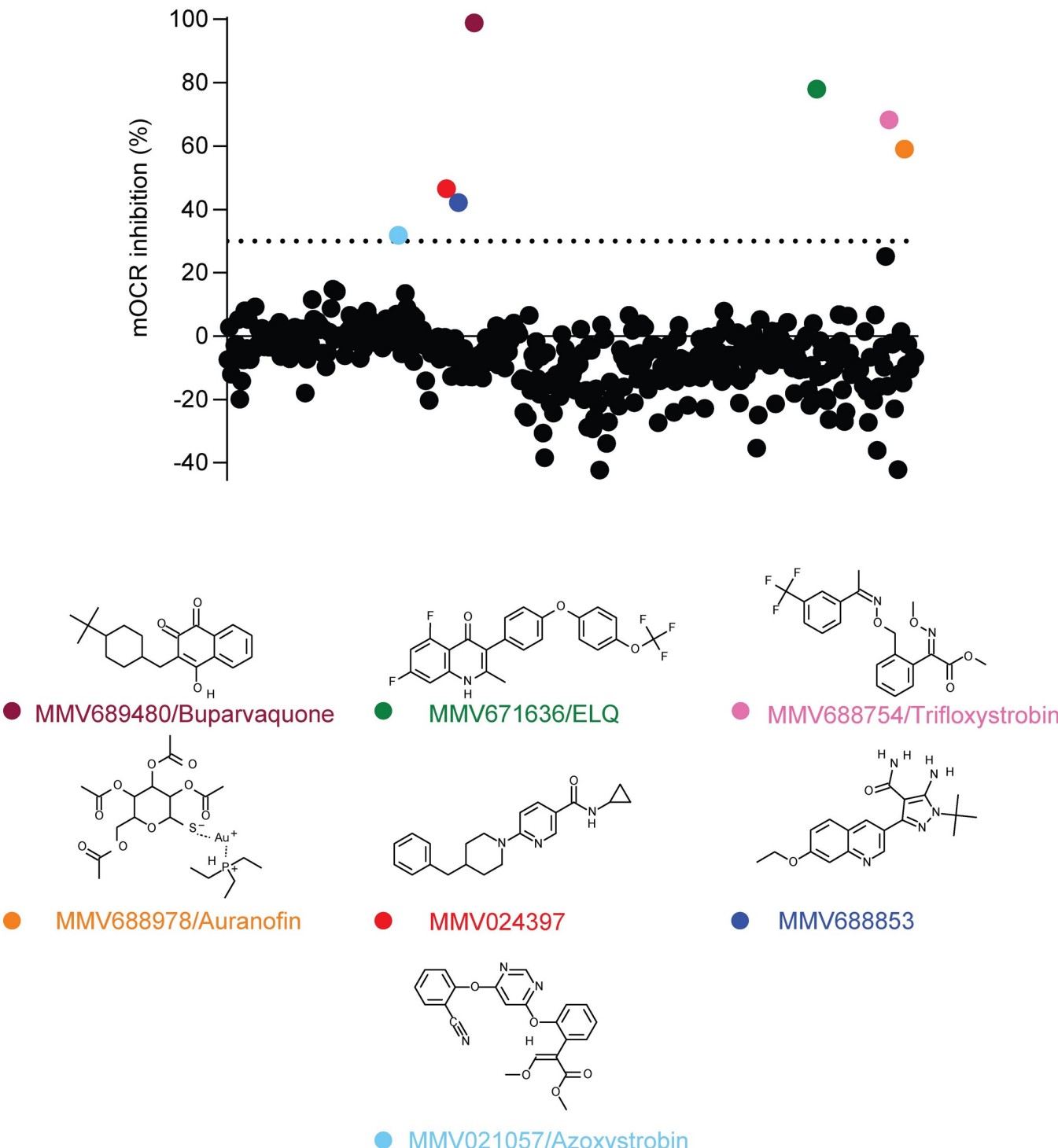

**Fig 1. Screening the MMV 'Pathogen Box' for inhibitors of O₂ consumption in *T. gondii*.** The O₂ consumption rate (OCR) of extracellular *T. gondii* parasites was measured in a 96-well plate using a Seahorse XFe96 extracellular flux analyzer. Compounds from the MMV 'Pathogen Box' were added to wells at a final concentration of 1 μM, and the change in OCR was monitored in real-time after each addition. Percent inhibition of OCR by each of the 400 compounds was calculated relative to complete inhibition observed after addition of the known OCR inhibitors atovaquone (1 μM) and antimycin A (10 μM), with each compound represented by a dot. A >30% inhibition cut off was applied (dotted line), with seven compounds inhibiting OCR by >30% at 1 μM (coloring of dots corresponds to coloring of labels of the chemical structures shown below). These hits included MMV689480/buparvaquone (burgundy), the endochin-like quinolone (ELQ) MMV671636 (green), MMV688754/trifloxystrobin (pink), MMV688978/auranofin (orange), MMV024397 (red), the aminopyrazole carboxamide MMV688853 (dark blue), and MMV021057/azoxystrobin (light blue). Data are from a single experiment, with the plate layouts and data points summarized in S1 Table.

**Table 1. Effects of the identified MMV Pathogen Box compounds on *T. gondii* proliferation.** Determination of the inhibitory properties of the identified compounds on the proliferation of (**A**) human foreskin fibroblast (HFF) cells or (**B**) *T. gondii* parasites, including wild type (WT) RH, WT ME49, atovaquone-resistant (ATV$^R$) ME49, and ELQ-300-resistant (ELQ$^R$) RH and the corresponding ELQ sensitive (ELQ$^S$) parental strains. Data are reported as average EC$_{50}$ (nM) ± SEM from three or more independent experiments. The selectivity index (SI) was calculated by dividing the EC$_{50}$ against HFF cells by the EC$_{50}$ against WT RH *T. gondii* parasites, with SI values >1 indicating increased selectivity towards the parasite. The fold change (FC) was calculated by dividing the EC$_{50}$ against ATV$^R$ or ELQ$^R$ parasites by the EC$_{50}$ against WT ME49 or ELQ$^S$ parasites respectively, with FC values >1 indicating increased resistance and FC values <1 indicating increased sensitivity to the tested compounds in the ATV$^R$ or ELQ$^R$ strains. Paired t-tests were performed to compare the EC$_{50}$ of the compounds in WT vs ATV$^R$ or ELQ$^S$ vs ELQ$^R$ parasites, and *p*-values are depicted as ns = not significant ($p > 0.05$), * $p < 0.05$, ** $p < 0.01$. ND = not determined.

| | (A) Human | | (B) *T. gondii* | | | | | | |
| | HFF cells | | WT RH | WT ME49 | ATV$^R$ ME49 | | ELQ$^S$ RH | ELQ$^R$ RH | |
| Compound | EC$_{50}$ (nM) | SI | EC$_{50}$ (nM) | EC$_{50}$ (nM) | EC$_{50}$ (nM) | FC | EC$_{50}$ (nM) | EC$_{50}$ (nM) | FC |
|---|---|---|---|---|---|---|---|---|---|
| Atovaquone | >10,000 | >961 | 10.4 ± 0.5 | 14 ± 4 | 284 ± 34 | 20 * | 17 ± 2 | 3.5 ± 1.2 | 0.2 ** |
| Buparvaquone | >10,000 | >14,000 | 0.7 ± 0.1 | 0.7 ± 0.2 | 163 ± 14 | 233 ** | 1.3 ± 0.4 | 0.4 ± 0.03 | 0.3 ns |
| MMV671636 | ND | ND | 3.0 ± 0.2 | ND | ND | ND | ND | ND | ND |
| Auranofin | 2,793 ± 914 | 27 | 102 ± 27 | 92 ± 13 | 191 ± 44 | 2 ns | 364 ± 119 | 299 ± 42 | 0.8 ns |
| Trifloxystrobin | >10,000 | >357 | 28 ± 2 | 67 ± 18 | 24 ± 5 | 0.4 ns | 134 ± 4 | 39 ± 5 | 0.3 ** |
| Azoxystrobin | >10,000 | >32 | 310 ± 32 | 579 ± 48 | 232 ± 36 | 0.4 * | 659 ± 58 | 291 ± 4 | 0.4 * |
| MMV024397 | >10,000 | >42 | 238 ± 30 | 153 ± 18 | 441 ± 92 | 2.9 ns | 259 ± 40 | 121 ± 20 | 0.5 * |
| MMV688853 | >10,000 | >145 | 69 ± 12 | 178 ± 14 | 133 ± 4 | 0.7 ns | 151 ± 13 | 290 ± 12 | 1.9 * |
| ELQ-300 | ND | ND | ND | ND | ND | ND | 39 ± 4 | 263 ± 28 | 6.7 * |
| Antimycin A | ND | ND | ND | ND | ND | ND | 77 ± 9 | 454 ± 86 | 5.9 ns |

dimeric Tomato (tdTomato) red fluorescent protein using a previously described fluorescence-based 96-well plate proliferation assay [39]. All seven compounds inhibited *T. gondii* proliferation with sub- to high-nanomolar EC$_{50}$ values, with buparvaquone (EC$_{50}$ ± SEM = 0.7 ± 0.1 nM, n = 3) and the ELQ MMV671636 (EC$_{50}$ ± SEM = 3.0 ± 0.2 nM, n = 3) the most potent, and azoxystrobin (EC$_{50}$ ± SEM = 310 ± 32 nM, n = 3) the least (Table 1; S1 Fig). Given that ELQ compounds are well-characterized ETC inhibitors [11,12], we did not include MMV671636 in further experiments.

We queried the extent to which the compounds identified in our screen exhibited selectivity for parasites over human cells. To test this, we measured the proliferation of human foreskin fibroblast (HFF) cells at a range of compound concentrations. Most of the compounds did not exhibit any effects on the proliferation of HFF cells at the highest concentration tested (10 μM), with the exception of auranofin (EC$_{50}$ ± SEM = 2,793 ± 914 nM, n = 3), which had a selectivity index (SI) of 27 for parasites over human cells (Table 1; S2 Fig). These data indicate that, in proliferation assays, all of the compounds identified in our screen exhibit considerable selectivity for *T. gondii* parasites over their hosts.

The ETC is a validated drug target in *P. falciparum* parasites [13], and we reasoned that the identified inhibitors of OCR in *T. gondii* may also act against the ETC of *P. falciparum*. We first tested whether the identified compounds could inhibit proliferation of the disease-causing asexual blood stage of 3D7 strain *P. falciparum*. Five of the six compounds inhibited *P. falciparum* proliferation, most with sub- to high-nanomolar EC$_{50}$ values (Table 2; Fig 2). While MMV688853 was an effective inhibitor of *T. gondii* proliferation, we found that it had little effect on the proliferation of *P. falciparum* at the concentration range we tested (up to 6.25 μM; Fig 2H). As an initial measure for whether they act specifically on the ETC of *P. falciparum* or whether they have broader cellular targets, we tested the ability of the identified compounds to inhibit the proliferation of yDHODH-expressing *P. falciparum* parasites, which are less dependent on the ETC for proliferation [9]. We observed that yDHODH-expressing parasites grew better than WT in the presence of four of the compounds (buparvaquone, trifloxystrobin, azoxystrobin and MMV024397) and the known ETC inhibitor atovaquone

**Table 2. Effects of the identified MMV Pathogen Box compounds on *P. falciparum* proliferation.** Determination of the inhibitory properties of the identified compounds on the proliferation of *P. falciparum* parasite strains, including WT 3D7, yeast dihydroorotate dehydrogenase (yDHODH)-expressing 3D7, and ATV$^R$ 3D7 and the equivalent ATV-sensitive parental WT 3D7 strain. As the yDHODH and ATV$^R$ strains were generated in different laboratories, proliferation of the WT 3D7 background strain of each was determined for comparisons. Data are reported as average EC$_{50}$ (nM) ± SEM from three or more independent experiments. The fold change (FC) was calculated by dividing the EC$_{50}$ against ATV$^R$ parasites by the EC$_{50}$ against WT parasites, with FC values >1 indicating increased resistance to the tested compounds in the ATV$^R$ strain. Paired t-tests were performed to compare the EC$_{50}$ of the compounds in WT vs ATV$^R$ parasites, and *p*-values are depicted as ns = not significant ($p > 0.05$), * $p < 0.05$, ** $p < 0.01$. ND = not determined.

| | *P. falciparum* | | | | |
| | **WT** | **yDHODH** | **WT** | **ATV$^R$** | |
| **Compound** | **EC$_{50}$ (nM)** | **EC$_{50}$ (nM)** | **EC$_{50}$ (nM)** | **EC$_{50}$ (nM)** | **FC** |
|---|---|---|---|---|---|
| Atovaquone | 0.13 ± 0.02 | > 10 | 0.31 ± 0.04 | 7.6 ± 0.9 | 24 ** |
| Chloroquine | 6.6 ± 1.3 | 7.70 ± 0.07 | 19 ± 2 | 19 ± 3 | 1 ns |
| Buparvaquone | 1.2 ± 0.3 | > 12.5 | 10.9 ± 1.2 | 1,160 ± 215 | 106 * |
| Auranofin | 2,040 ± 410 | 1,810 ± 490 | 2,831 ± 503 | 2,783 ± 362 | 1 ns |
| Trifloxystrobin | 44 ± 16 | > 250 | 33 ± 7 | 131 ± 12 | 4 * |
| Azoxystrobin | 23 ± 9 | > 125 | 12 ± 1 | 31 ± 7 | 2.6 ns |
| MMV024397 | 308 ± 18 | 3,740 ± 1,280 | 400 ± 48 | 602 ± 93 | 1.5 * |
| MMV688853 | > 6,250 | > 6,250 | >40,000 | >40,000 | ND |

(Table 2; Fig 2), consistent with these compounds acting primarily on the ETC in *P. falciparum*. By contrast, yDHODH and WT parasites were equally inhibited in the presence of auranofin and the control compound chloroquine, which does not target the ETC (Table 2; Fig 2). This observation suggests that auranofin perturbs parasite proliferation independently of ETC inhibition. Together, these results indicate that most of the identified compounds are selective inhibitors of the ETC in *P. falciparum*.

To explore their potency at inhibiting OCR in *T. gondii*, we investigated the effects of a range of concentrations of each compound on parasite OCR using the Seahorse XFe96 flux analyzer. All compounds inhibited the OCR of *T. gondii* tachyzoites in a dose-dependent manner (Table 3; S3C-S3I Fig). Most of the tested compounds showed rapid inhibition of OCR at the higher concentrations tested (as shown for atovaquone, S3A Fig). By contrast, inhibition of OCR by auranofin occurred more gradually over time, even at the highest concentration tested (S3B Fig), suggesting that the effects of auranofin on OCR may occur in a different manner to the other identified compounds.

We examined whether there was a correlation between the potency of compounds at inhibiting *T. gondii* OCR compared to their ability to inhibit parasite proliferation. We generally found that the most potent OCR inhibitors were also the most potent inhibitors of parasite proliferation (*e.g.* atovaquone and trifloxystobin), whereas the least potent OCR inhibitors were also the least potent inhibitors of parasite proliferation (*e.g.* azoxystrobin; S3J Fig). An exception was buparvaquone, which was the most potent proliferation inhibitor of the compounds tested but was a less potent OCR inhibitor than compounds like atovaquone and trifloxystrobin. EC$_{50}$ values against OCR were higher than EC$_{50}$ values against parasite proliferation (S3J Fig), possibly reflecting differences in the temporal scales of the OCR and proliferation assays (OCR readings were in real-time, whereas proliferation was calculated across several days) and the different nature of the biological processes being measured (*e.g.* partial inhibition of the ETC might lead to a proportionally greater decrease in parasite proliferation).

In addition to measuring OCR, the Seahorse XFe96 extracellular flux analyzer simultaneously measures the extracellular acidification rate (ECAR), which provides a general measure of parasite metabolic activity [27,29]. We observed that most of the test compounds

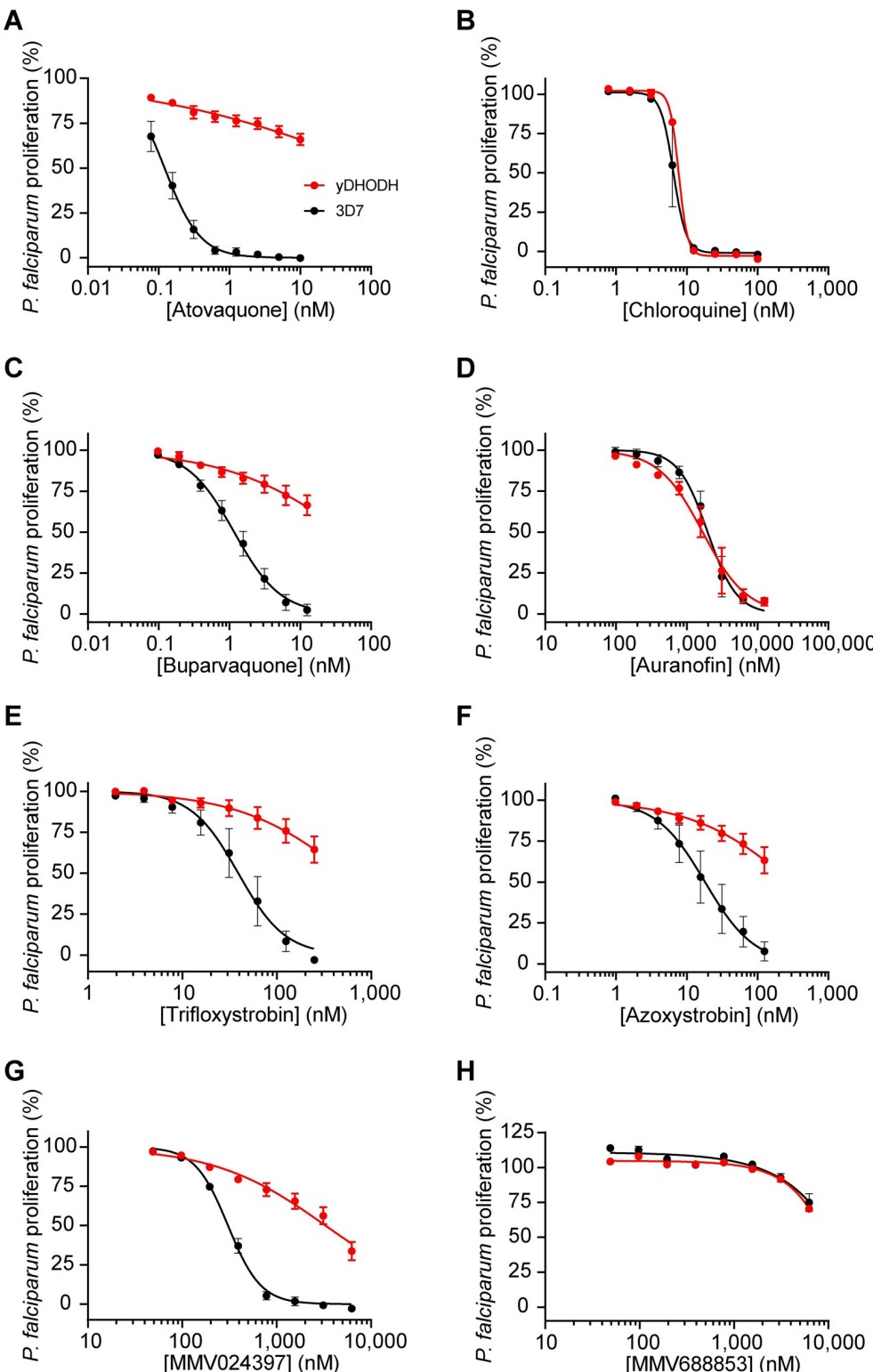

**Fig 2. Identification of selective inhibitors of the ETC in *P. falciparum*.** Dose-response curves depicting the proliferation of WT (black) or yeast dihydroorotate dehydrogenase (yDHODH)-expressing (red) *P. falciparum* parasites in the presence of increasing concentrations of (**A**) the known ETC inhibitor atovaquone, (**B**) chloroquine, a compound that does not inhibit the ETC, (**C**) buparvaquone, (**D**) auranofin, (**E**) trifloxystrobin, (**F**) azoxystrobin, (**G**) MMV024397, or (**H**) MMV688853 after 96 h of culture. Values are expressed as a percentage of the average

proliferation of the drug-free control, and represent the mean ± SEM of three independent experiments performed in triplicate; error bars that are not visible are smaller than the symbol.

inhibited OCR in *T. gondii* without significantly inhibiting ECAR (Figs 3A and S4), suggesting that they selectively target the ETC of the parasite. By contrast, treatment with auranofin resulted in a concomitant and significant decrease in both OCR and ECAR (Figs 3A and S4). This provides additional evidence that auranofin acts in a different manner to the other identified compounds. To explore this further, we assessed the viability of parasites upon auranofin treatment. We treated *T. gondii* parasites with 1, 20 or 100 μM auranofin, or 10 μM atovaquone as a control, stained parasites with propidium iodide (PI, a fluorescent dye that enters non-viable cells), and quantified parasite viability by flow cytometry (S5 Fig). We observed that treatment with auranofin led to a rapid, dose-dependent decrease in parasite viability over the 140-minute time course of the assay (Fig 3B). By contrast, treatment with the selective ETC inhibitor atovaquone caused minimal loss of parasite viability within this timeframe (Fig 3B), suggesting the decreased viability observed upon auranofin treatment was not due to ETC inhibition. These data suggest that auranofin is not a selective inhibitor of the ETC but instead perturbs broader parasite functions, resulting in a decrease in parasite viability and a secondary impairment of ETC activity.

We conclude that most of the compounds identified in our initial screen inhibit the proliferation of *T. gondii* and *P. falciparum* parasites, and act selectively on the ETC of these parasites. A strength of the Seahorse XFe96 flux analyzer-based screening approach is its ability to simultaneously measure OCR and ECAR, and thereby enable the differentiation of compounds that directly inhibit the ETC from those–such as auranofin–that have a broader effect on parasite metabolism or viability.

## MMV688853 inhibits the ETC in a *Tg*CDPK1-independent manner

One of the hit compounds identified in our ETC inhibitor screen was the aminopyrazole carboxamide scaffold compound MMV688853, which has been reported previously to be an inhibitor of *T. gondii* calcium-dependent protein kinase 1 (*Tg*CDPK1) [33,34]. *Tg*CDPK1 is a cytosolic protein that has been shown to be critical for parasite invasion of host cells [40]. We hypothesized that either *Tg*CDPK1 has an additional role in the ETC or that MMV688853 has a second target in these parasites. *Tg*CDPK1 has a glycine residue at the mouth of the pocket

**Table 3. Inhibitory activities of MMV Pathogen Box compounds against O$_2$ consumption rate in *T. gondii* and *P. falciparum*.** Determination of the O$_2$ consumption rate (OCR) inhibitory properties of the identified compounds on (**A**) WT RH strain *T. gondii* parasites, and on (**B**) WT 3D7 strain *P. falciparum* parasites, using a Seahorse XFe96 flux analyzer. *T. gondii* experiments were conducted on intact parasites, and *P. falciparum* experiments measured malate-dependent OCR in digitonin-permeabilized parasites. Data are reported as average EC$_{50}$ value against OCR (EC$_{50}^{OCR}$) (μM) ± SEM from three or more independent experiments. ND = not determined.

| | (A) *T. gondii* OCR | (B) *P. falciparum* OCR |
|---|---|---|
| Compound | EC$_{50}^{OCR}$ (μM) | EC$_{50}^{OCR}$ (μM) |
| Atovaquone | 0.18 ± 0.05 | 0.022 ± 0.008 |
| Buparvaquone | 1.18 ± 0.69 | ND |
| Auranofin | 2.48 ± 0.46 | >100 |
| Trifloxystrobin | 0.50 ± 0.02 | 0.042 ± 0.017 |
| Azoxystrobin | 7.05 ± 3.08 | 0.015 ± 0.002 |
| MMV024397 | 2.81 ± 0.66 | 0.413 ± 0.051 |
| MMV688853 | 2.76 ± 0.48 | >10 |

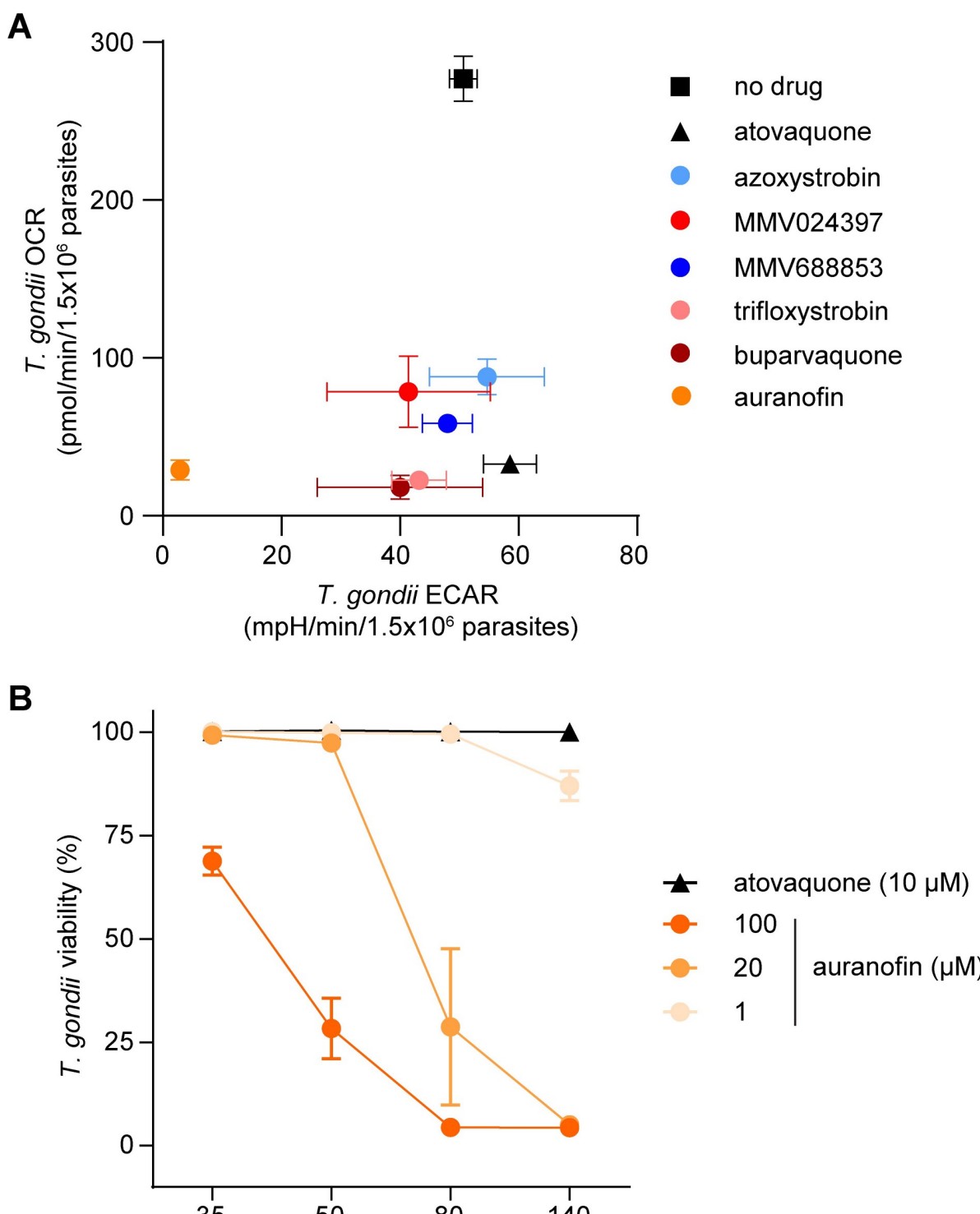

**Fig 3. Identification of selective and off-target inhibitors of the ETC in *T. gondii* parasites. (A)** $O_2$ consumption rate (OCR) versus extracellular acidification rate (ECAR) of *T. gondii* parasites treated with either no drug (black square), atovaquone (black triangle; 10 µM), azoxystrobin (light blue; 80 µM), MMV024397 (red; 20 µM), MMV688853 (dark blue; 20 µM), trifloxystrobin (pink; 10 µM), buparvaquone (burgundy; 20 µM) or auranofin (orange; 80 µM) assessed using a Seahorse XFe96 flux analyzer. Data represent the mean OCR and ECAR ± SEM of three independent experiments, and are derived from the top concentration of inhibitor tested in S3 Fig. Statistical analyses

of these data are presented in S4 Fig. **(B)** Viability of extracellular *T. gondii* parasites treated with atovaquone (black triangles, 10 μM) or auranofin (orange circles, 1–100 μM) for 35–140 minutes. Viability was assessed by flow cytometry of propidium iodide-stained parasites and normalized to a DMSO-treated vehicle control, with the gating strategy outlined in S5 Fig. Data represent the mean ± SEM of three independent experiments; error bars that are not visible are smaller than the symbol.

where MMV688853 and other *Tg*CDPK1 inhibitors bind (Fig 4A). Mutation of this so-called 'gatekeeper' residue to a bulky amino acid like methionine renders *Tg*CDPK1 resistant to inhibition by aminopyrazole carboxamide scaffold compounds like MMV688853 [34], as well as to pyrazolopyrimidine scaffold compounds such as 3-MB-PP1 [40] (Fig 4A). To test our hypotheses, we generated a tdTomato$^+$ *T. gondii* strain wherein the gatekeeper glycine residue at position 128 of *Tg*CDPK1 was mutated to methionine (*Tg*CDPK1$^{G128M}$; Fig 4A).

*Tg*CDPK1 is an important regulator of parasite invasion [40], a critical step in the lytic cycle of the parasite. Previous studies have shown that *Tg*CDPK1 inhibitors impair host cell invasion by WT but not *Tg*CDPK1$^{G128M}$ parasites [40]. To validate this, we tested the ability of MMV688853 to inhibit the invasion of WT and *Tg*CDPK1$^{G128M}$ parasites. While invasion of WT parasites was significantly inhibited by both MMV688853 and the control *Tg*CDPK1 inhibitor 3-MB-PP1, *Tg*CDPK1$^{G128M}$ parasites were able to invade in the presence of either compound (Fig 4B). By comparison, the ETC inhibitor atovaquone did not inhibit the invasion of either parasite strain (Fig 4B). These results indicate that MMV688853 inhibits *T. gondii* invasion in a *Tg*CDPK1-dependent manner.

We next tested the ability of MMV688853 to inhibit intracellular proliferation of WT and *Tg*CDPK1$^{G128M}$ parasites. We allowed parasites to invade host cells in the absence of inhibitors, then grew parasites for 19 h in the presence of MMV688853 or various control inhibitors and quantified the number of parasites per vacuole. MMV688853 inhibited intracellular proliferation of both WT and *Tg*CDPK1$^{G128M}$ parasites, with most vacuoles having only a single parasite (Fig 4C). Treatment with atovaquone resulted in similar impairment of intracellular proliferation (Fig 4C), with the majority of vacuoles containing 1–2 parasites. These data indicate that MMV688853 can inhibit intracellular proliferation independently of *Tg*CDPK1. Unexpectedly, the majority of both WT and *Tg*CDPK1$^{G128M}$ parasites grown in the presence of the control *Tg*CDPK1 inhibitor 3-MB-PP1 exhibited abnormal morphology (defined as vacuoles that contained misshapen parasites, possibly resulting from defects in cell division; Figs 4C and S6), suggesting an additional off-target effect of 3-MB-PP1.

As a test for whether the inhibition of $O_2$ consumption by MMV688853 occurs through inhibition of *Tg*CDPK1, we assessed the OCR of intact WT and *Tg*CDPK1$^{G128M}$ parasites after addition of increasing concentrations of MMV688853. We observed similar EC$_{50}$ values against OCR (EC$_{50}$$^{OCR}$) in WT and *Tg*CDPK1$^{G128M}$ parasites (Fig 4D). We also examined the ability of the alternative *Tg*CDPK1 inhibitor 3-MB-PP1 to inhibit OCR of WT and *Tg*CDPK1$^{G128M}$ parasites. In contrast to atovaquone and MMV688853, 3-MB-PP1 did not inhibit OCR in either WT or *Tg*CDPK1$^{G128M}$ parasites (Fig 4E). Together, these data indicate that MMV688853 acts on the ETC independently of *Tg*CDPK1, and that *Tg*CDPK1 does not have a role in the ETC.

Finally, we measured the effects of MMV688853 on the overall proliferation of WT and *Tg*CDPK1$^{G128M}$ *T. gondii* parasites through the lytic cycle. We measured parasite proliferation in the presence of increasing concentrations of MMV688853 over six days using a fluorescence proliferation assay. We observed a small (~two-fold) but not significant ($p = 0.076$) increase in the EC$_{50}$ of *Tg*CDPK1$^{G128M}$ parasites compared to the WT control (Fig 4F). Taken together, our data indicate that while MMV688853 inhibits parasite invasion by targeting *Tg*CDPK1 (Fig 4B), MMV688853 also has a second target in the ETC of the parasite (Fig 4D and 4E), and

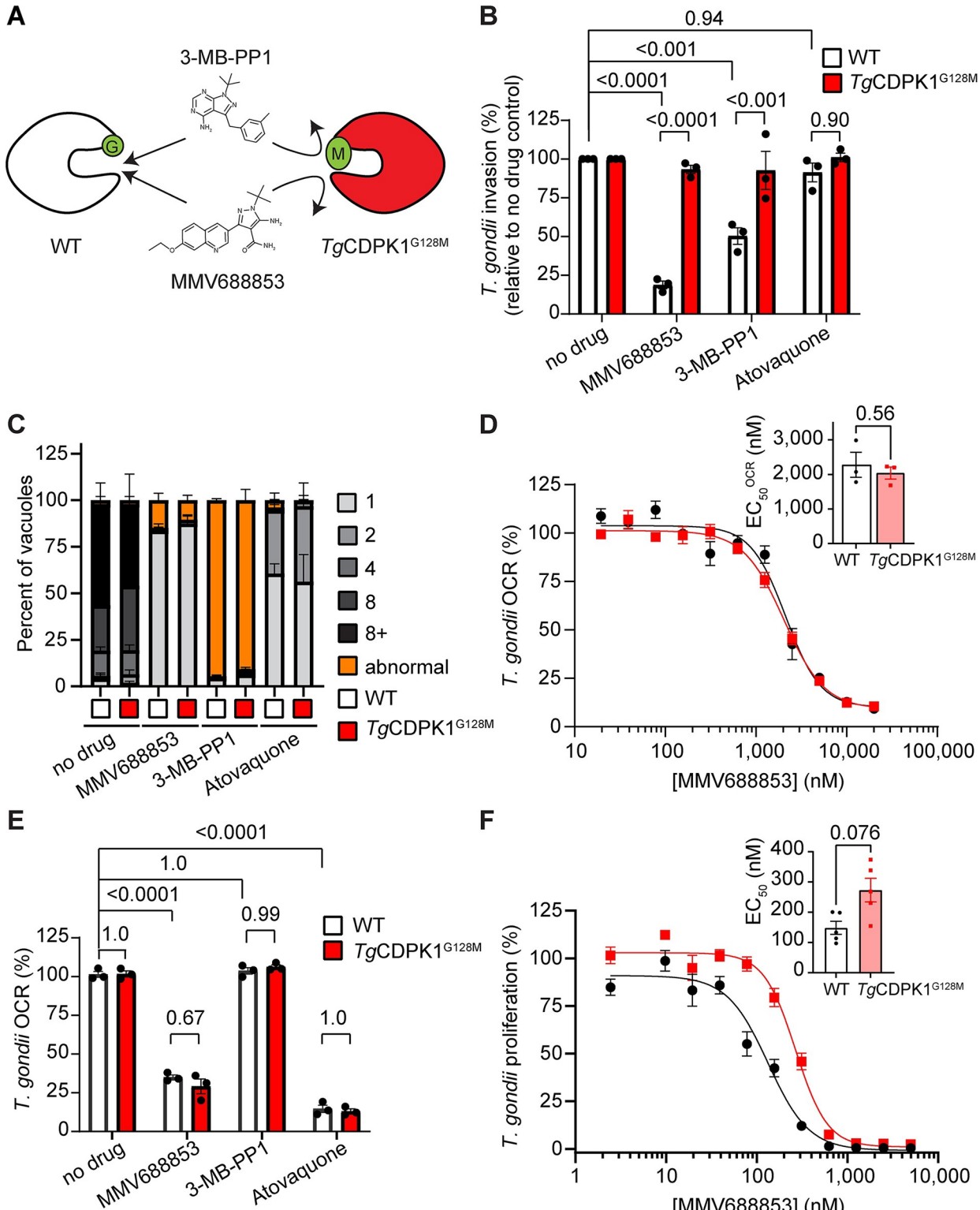

**Fig 4. MMV688853 dually targets *Tg*CDPK1 and the ETC in *T. gondii* parasites.** (A) Schematic depicting the small glycine gatekeeper residue of WT *Tg*CDPK1 (white) which enables inhibition by 3-MB-PP1 and MMV688853. Mutation of this residue to a larger methionine residue (*Tg*CDPK1$^{G128M}$, red) blocks inhibitor access to the binding site and thereby confers resistance to these compounds. **(B)** Percent invasion of parasites expressing WT *Tg*CDPK1 (white) or *Tg*CDPK1$^{G128M}$ (red) into host cells in the absence of drug (DMSO vehicle control), or the presence of MMV688853 (5 μM), 3-MB-PP1 (5 μM) or atovaquone (1 μM), normalized relative to the no-drug control. At least 100 parasites were counted

per experiment, with data representing the mean ± SEM of three independent experiments. ANOVA followed by Tukey's multiple comparisons test was performed with relevant $p$-values shown. (**C**) Intracellular proliferation assays depicting the percent of vacuoles containing 1–8+ (gray tones) or abnormal (orange) parasites when parasites expressing WT $Tg$CDPK1 (white) or $Tg$CDPK1$^{G128M}$ (red) were cultured in the absence of drug (DMSO vehicle control), or the presence of MMV688853 (5 μM), 3-MB-PP1 (5 μM) or atovaquone (1 μM) for 20 h. Abnormal morphology was defined as vacuoles that contained misshapen parasites (representative images in S6 Fig). At least 100 vacuoles were counted per condition, with data representing the mean ± SEM of three independent experiments. (**D**) Dose-response curves depicting the O$_2$ consumption rate (OCR) of parasites expressing WT $Tg$CDPK1 (black) or $Tg$CDPK1$^{G128M}$ (red) incubated with increasing concentrations of MMV688853 as a percentage of a no-drug (DMSO vehicle) control. Data represent the mean ± SEM of three independent experiments. Inset bar graph depicts the EC$_{50}^{OCR}$ ± SEM (nM) of three independent experiments. The $p$-value from a paired t-test is shown. (**E**) OCR of parasites expressing WT $Tg$CDPK1 (white) or $Tg$CDPK1$^{G128M}$ (red) incubated in the absence of drug (DMSO vehicle control), or in the presence of MMV688853 (5 μM), 3-MB-PP1 (5 μM) or atovaquone (1 μM), expressed as a percentage of the OCR prior to addition of compounds. Data represent the mean ± SEM of three independent experiments. ANOVA followed by Tukey's multiple comparisons test was performed with relevant $p$-values shown. (**F**) Dose-response curves depicting the percentage proliferation of parasites expressing WT $Tg$CDPK1 (black) or $Tg$CDPK1$^{G128M}$ (red) in the presence of increasing concentrations of MMV688853 over 6 days. Values are expressed as a percent of the average fluorescence from the no-drug control at mid-log phase growth in the fluorescence proliferation assay, and represent the mean ± SEM of five independent experiments conducted in triplicate; error bars that are not visible are smaller than the symbol. Inset bar graph depicts the EC$_{50}$ ± SEM (nM) of five independent experiments. The $p$-value from a paired t-test is shown.

this second target likely contributes to impairment of intracellular proliferation of the parasite by this compound (Fig 4F). Our study therefore identifies the ETC as a target of MMV688853 and potentially other aminopyrazole carboxamide scaffold compounds.

## Defining the targets of the candidate ETC inhibitors in *T. gondii* and *P. falciparum*

Having characterized the inhibitory properties of the candidate ETC inhibitors, we next sought to identify which component of the ETC these compounds target. To do this, we utilized a Seahorse XFe96 analyzer-based assay that we developed previously to pinpoint where a defect in the *T. gondii* ETC is occurring [28,29] (Fig 5A). Briefly, *T. gondii* parasites were starved for 1 h to deplete endogenous substrates. The parasite plasma membrane was permeabilized using a low concentration of the detergent digitonin, and parasites were incubated with one of two substrates that independently feed electrons to CoQ in the mitochondrion: 1) malate, which donates electrons to the ETC via a reaction catalyzed by the TCA cycle enzyme malate:quinone oxidoreductase; or 2) glycerol 3-phosphate (G3P), which donates electrons to the ETC independently of the TCA cycle via a reaction catalyzed by G3P dehydrogenase. Following substrate addition, the candidate inhibitor was added at a concentration that we previously showed maximally inhibited OCR (S3 Fig) and the change in OCR was measured. If OCR elicited by both substrates was inhibited, this provided evidence that the inhibitor was acting downstream of CoQ (*i.e.* on ETC Complexes III or IV; Fig 5A; [29]). To differentiate between Complex III and Complex IV inhibition, samples were next treated with the substrate $N,N,N'$, $N'$-tetramethyl-$p$-phenylenediamine dihydrochloride (TMPD), which donates electrons directly to CytC and consequently bypasses Complex III (Fig 5A). If inhibition of OCR was rescued by addition of TMPD, this provided evidence that the inhibitor was acting upstream of CytC (*e.g.* on ETC Complex III). Finally, samples were treated with the Complex IV inhibitor sodium azide (NaN$_3$) to validate that the observed TMPD-dependent OCR was a result of Complex IV activity.

We observed that all compounds significantly inhibited OCR regardless of whether the parasites were utilizing malate or G3P as ETC substrates (Figs 5B–5H and S7), suggesting that inhibition by these compounds was occurring downstream of CoQ. While most compounds inhibited OCR almost immediately, auranofin inhibition was more gradual (Fig 5D), consistent with our previous evidence of indirect inhibition of the ETC by this compound (Fig 3). Furthermore, OCR could be rescued by TMPD for all compounds except auranofin (Figs 5B– 5H and S7), which indicates that these compounds inhibit upstream of CytC. Together, these

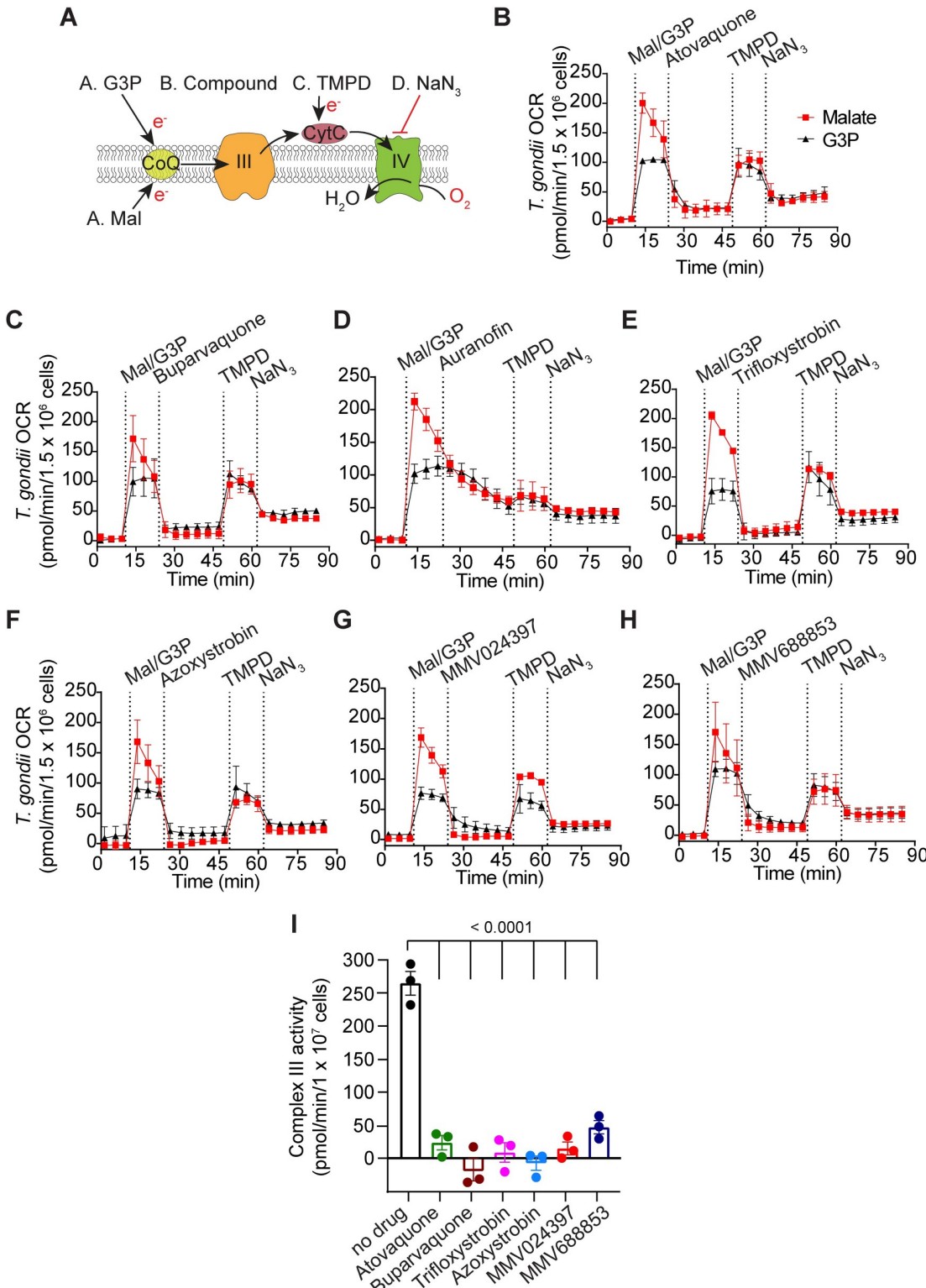

**Fig 5. An assay to characterize the targets of the candidate ETC inhibitors identifies chemically diverse Complex III inhibitors.** (A) Schematic of the assay measuring the $O_2$ consumption rate (OCR) of plasma membrane-permeabilized *T. gondii* parasites. Parasites were starved for 1 hour to deplete endogenous substrates then permeabilized with digitonin before the addition of the following substrates and inhibitors: Port A, the substrates malate (Mal) or glycerol 3-phosphate (G3P); Port B, the test compound; Port C, TMPD; Port D, sodium azide ($NaN_3$). CoQ, coenzyme Q; III, Complex III; CytC, cytochrome *c*; IV, Complex

IV; e⁻, electrons. **(B-H)** Traces depicting parasite OCR over time when supplying Mal (red squares) or G3P (black triangles) as a substrate. The candidate ETC inhibitors were **(B)** atovaquone (1.25 μM), **(C)** buparvaquone (5 μM), **(D)** auranofin (10 μM), **(E)** trifloxystrobin (2.5 μM), **(F)** azoxystrobin (80 μM), **(G)** MMV024397 (20 μM), or **(H)** MMV688853 (20 μM). Values represent the mean ± SD of three technical replicates and are representative of three independent experiments; error bars that are not visible are smaller than the symbol. Dotted lines represent the time points of each injection, and data points for each condition have been connected by lines to aid interpretation. Quantifications of the data in (B-H) are presented in S7 Fig. **(I)** *T. gondii* Complex III enzymatic activity was assessed in the presence of DMSO (no-drug), atovaquone (1.25 μM), buparvaquone (5 μM), trifloxystrobin (2.5 μM), azoxystrobin (80 μM), MMV024397 (20 μM) or MMV688853 (20 μM). Data represent the mean ± SEM of three independent experiments each conducted in duplicate. ANOVA followed by Dunnett's multiple comparisons test were performed and *p*-values are shown.

data indicate that the on-target compounds identified in our screen all act via inhibition of ETC Complex III.

To validate these results, we performed a direct, spectrophotometric-based Complex III enzymatic assay on parasite extracts in the absence or presence of inhibitors. We observed that Complex III activity was significantly lower in the presence of all tested on-target inhibitors than in the no-drug control (Figs 5I and S8), indicating that the identified compounds are indeed Complex III inhibitors. The inhibitory activity of auranofin could not be assessed via this assay since we observed apparent enzyme activity upon auranofin addition even in the absence of parasite extract (S8A Fig).

To begin to define the targets of the identified compounds in *P. falciparum*, we tested the ability of the compounds to inhibit OCR in permeabilized *P. falciparum* parasites that were supplied malate as a substrate (Fig 6). We observed that all compounds except auranofin (Fig 6C) and MMV688853 (Fig 6G) could inhibit OCR of *P. falciparum* parasites, and that TMPD restored OCR in all cases (Fig 6B–6G). These results are consistent with most of the compounds that inhibited Complex III in *T. gondii* inhibiting the same complex in *P. falciparum*, although our assay cannot rule out that they target malate oxidation instead.

To investigate the potency of each compound in inhibiting OCR of *P. falciparum*, we performed a dose-response experiment (S9 Fig). All compounds except MMV688853 (S9F Fig) and auranofin (S9B Fig) inhibited OCR of digitonin-permeabilized *P. falciparum* in a dose-dependent manner, with $EC_{50}^{OCR}$ values in the sub-micromolar range (Table 3). These $EC_{50}^{OCR}$ values were lower than the $EC_{50}^{OCR}$ values observed in *T. gondii* (Table 3), which may be due to the *P. falciparum* $EC_{50}^{OCR}$ values being determined in digitonin-permeabilized parasites and those for *T. gondii* in intact extracellular parasites. Together, our data provide evidence that most identified compounds are potent inhibitors of the ETC in both *T. gondii* and *P. falciparum*, and that these compounds target Complex III. Our data also point to some differences in the activity of these compounds between *T. gondii* and *P. falciparum*, most notably with MMV688853, which inhibits Complex III in *T. gondii*, but is inactive against the ETC in *P. falciparum* at the concentrations tested.

## Atovaquone-resistant *T. gondii* and *P. falciparum* exhibit limited cross-resistance to most of the identified MMV compounds

Atovaquone resistance is known to arise rapidly in apicomplexans, both in the field and the laboratory [19,41,42]. Atovaquone acts by binding the CoQ oxidation (Qₒ) site of Complex III, which is in a pocket formed in part by the cytochrome *b* protein of Complex III. Mutations in Qₒ site residues of cytochrome *b*, a protein encoded on the mitochondrial genome of apicomplexan parasites, confer varying degrees of atovaquone resistance in both *T. gondii* and *Plasmodium spp.* [19,20,43]. We tested whether atovaquone-resistant strains of *T. gondii* and *P. falciparum* parasites exhibited cross-resistance to any of the Complex III inhibitors identified in our screen.

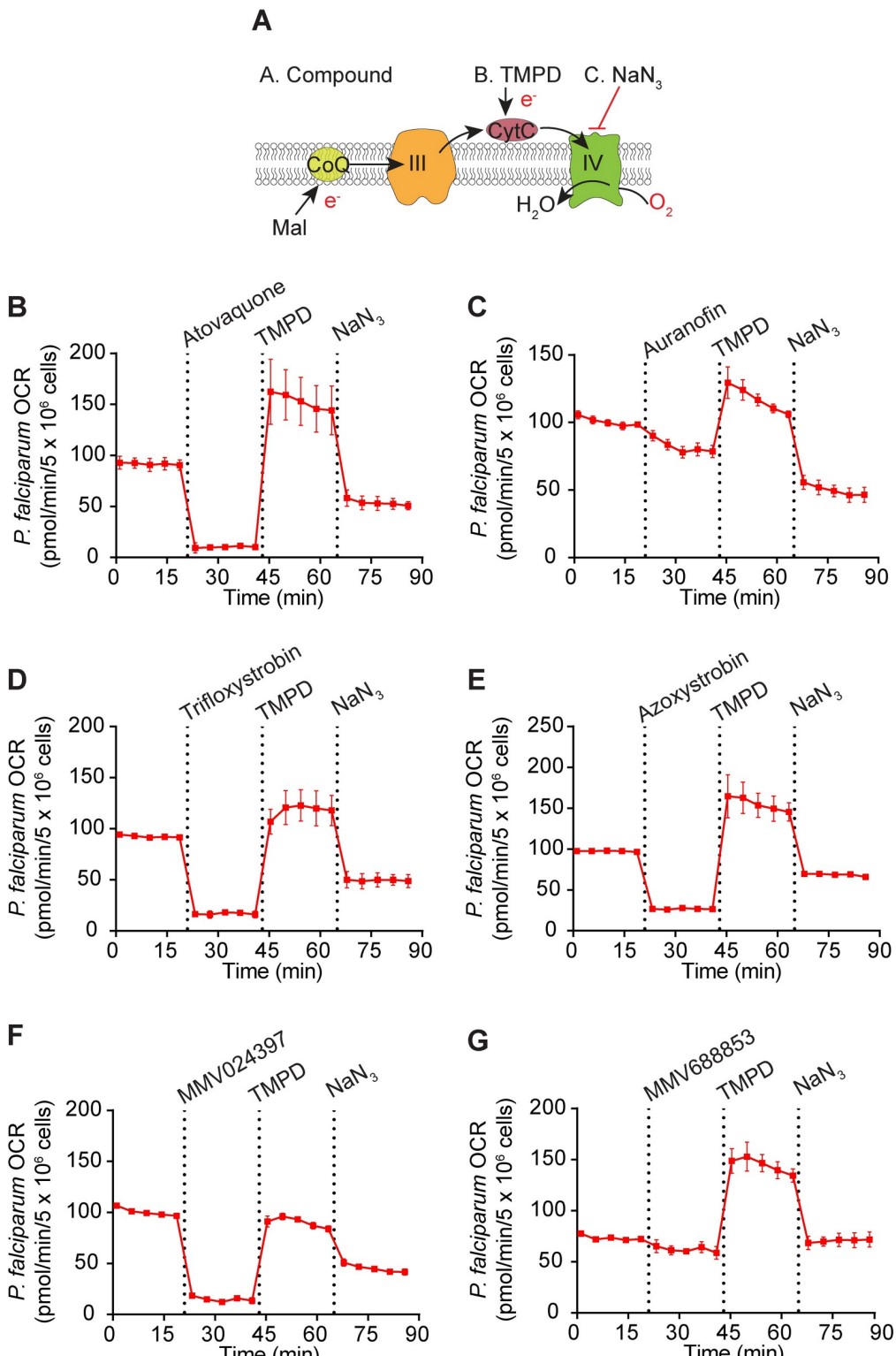

**Fig 6. Most of the candidate ETC inhibitors target the ETC upstream of cytochrome *c* in *P. falciparum* parasites.** (**A**) Schematic of the assay measuring the $O_2$ consumption rate (OCR) of permeabilized *P. falciparum* parasites supplied malate (Mal) as a substrate. The following addition of substrates and inhibitors were performed: Port A, the test compound; Port B, TMPD; Port C, sodium azide ($NaN_3$). CoQ, coenzyme Q; III, Complex III; CytC, cytochrome *c*; IV, Complex IV; e⁻, electrons. (**B-G**) Traces depicting parasite OCR over time when supplying Mal as a substrate.

The candidate ETC inhibitors tested (all at 10 µM) were **(B)** atovaquone, **(C)** auranofin, **(D)** trifloxystrobin, **(E)** azoxystrobin, **(F)** MMV024397, or **(G)** MMV688853. Values represent the mean ± SD of three technical replicates and are representative of three independent experiments; error bars that are not visible are smaller than the symbol. Dotted lines represent the time points of each injection, and data points for each condition have been connected by lines to aid interpretation.

We first tested the effects of the identified inhibitors on a previously described atovaquone-resistant (ATV$^R$) ME49 strain of *T. gondii* which has an isoleucine to leucine substitution at position 254 (I254L) of cytochrome *b* [19]. We integrated tdTomato into WT ME49 and ATV$^R$ *T. gondii* parasites and performed fluorescence proliferation assays to compare the ability of atovaquone and the test compounds to inhibit proliferation of these two strains. As demonstrated for RH parasites (Table 1), all six compounds inhibited WT ME49 *T. gondii* proliferation at sub-micromolar concentrations (Table 1; Fig 7). As expected, the ATV$^R$ strain was resistant to atovaquone, with a ~20-fold higher EC$_{50}$ than WT parasites ($p = 0.017$; Table 1; Fig 7A). ATV$^R$ parasites were cross-resistant to buparvaquone (~233-fold, $p = 0.0076$; Table 1; Fig 7B). Interestingly, ATV$^R$ *T. gondii* parasites were slightly sensitized to the antifungal strobilurin family compounds azoxystrobin (~2.5-fold, $p = 0.025$; Table 1; Fig 7E) and trifloxystrobin (~2.8-fold, $p = 0.077$; Table 1; Fig 7D), and showed minimal cross-resistance against the other tested inhibitors.

We next tested whether an atovaquone resistance-conferring mutation in *P. falciparum* would result in similar changes in sensitivity to the inhibitors identified from our screen. We generated an atovaquone-resistant (ATV$^R$) *P. falciparum* parasite strain by drug pressure which had a valine to leucine substitution at position 259 (V259L) in cytochrome *b*, and compared their proliferation in the presence of the candidate ETC inhibitors to WT parasites (Fig 8). As expected, the ATV$^R$ *P. falciparum* strain was resistant to atovaquone, with a ~24-fold higher EC$_{50}$ than WT parasites ($p = 0.0087$; Table 2; Fig 8A), but not to chloroquine (Table 2; Fig 8B). Like in *T. gondii*, we observed cross-resistance to buparvaquone (~106-fold, $p = 0.017$; Table 2; Fig 8C). We observed little to no cross-resistance of ATV$^R$ *P. falciparum* parasites to auranofin (no change; Table 2; Fig 8D), trifloxystrobin (~4-fold, $p = 0.012$; Table 2; Fig 8E), azoxystrobin (~1.5 fold, $p = 0.055$; Table 2; Fig 8F), or MMV024397 (~1.5 fold, $p = 0.028$; Table 2; Fig 8G). MMV688853 exhibited minimal inhibition of parasite proliferation in the ATV$^R$ strain even at the highest concentration tested (40 µM; Table 2; Fig 8H), consistent with the previous assays with WT *P. falciparum* (Fig 2H). Together, these data indicate that ATV$^R$ parasites do not exhibit a great degree of cross-resistance to most of our compounds (with the exception of buparvaquone, which belongs to the same hydroxy-naphthoquinone class as atovaquone).

## ELQ-300-resistant *T. gondii* parasites exhibit no cross-resistance to most of the identified MMV compounds

Endochin-like quinolones (ELQs) represent another class of anti-apicomplexan compounds that target Complex III. In contrast to atovaquone, many ELQs, including the commercially available ELQ-300, target the CoQ reduction (Q$_i$) site of Complex III, and mutations in the Q$_i$ site can confer resistance to these ELQs [12,44,45]. We set out to determine whether ELQ-resistant *T. gondii* parasites exhibited cross-resistance to the inhibitors identified in our screen. To test this, we utilized an existing *T. gondii* parasite strain containing a threonine to proline mutation at residue 222 in the Q$_i$ site of cytochrome *b* (T222P) that confers resistance to ELQ-300 and other Q$_i$ site targeting compounds [44,45]. We introduced a tdTomato transgene into the genomes of the ELQ resistant (ELQ$^R$) strain and corresponding ELQ-sensitive (ELQ$^S$)

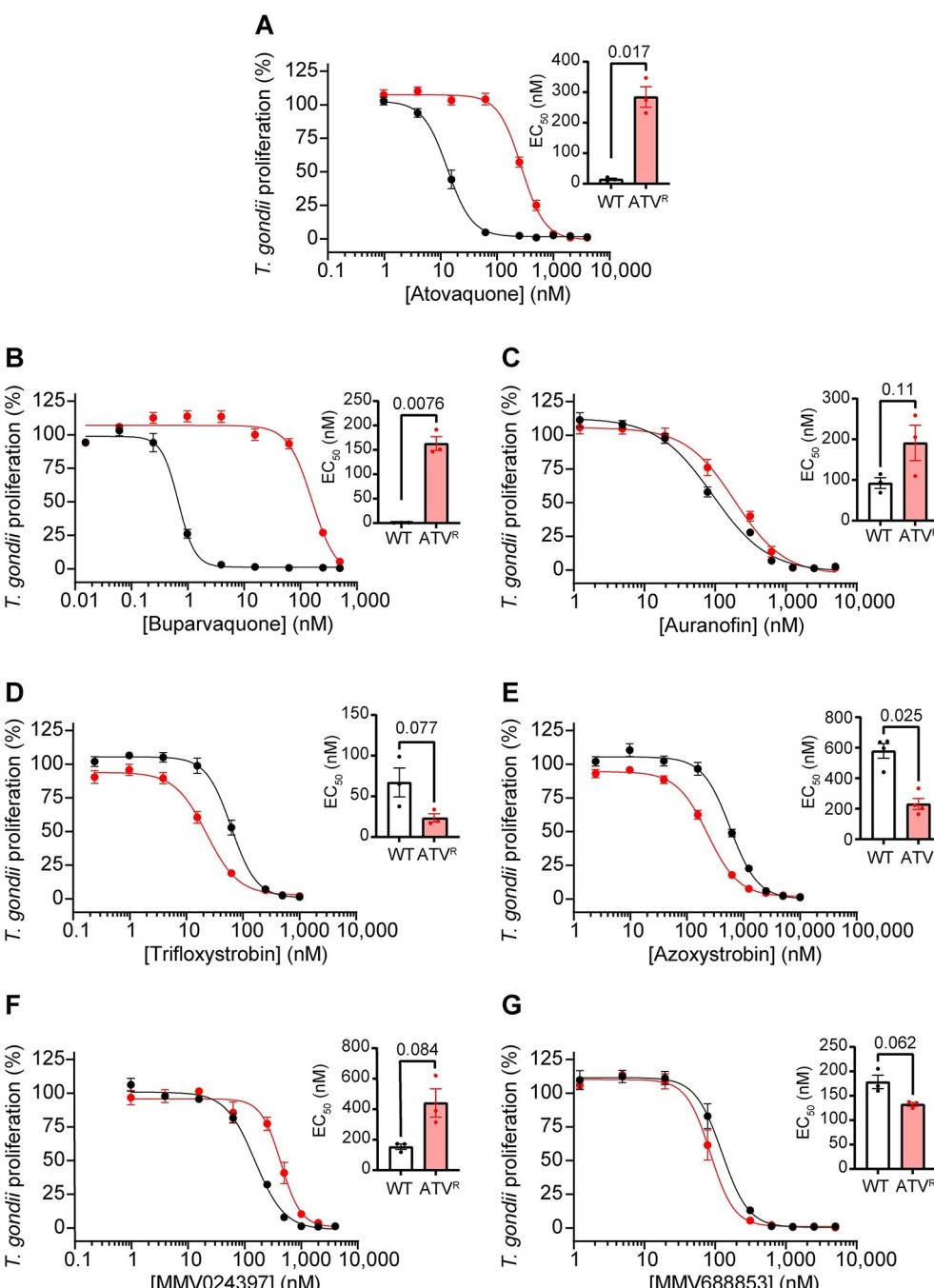

**Fig 7. Assessing the activity of ETC inhibitors against atovaquone-resistant *T. gondii* parasites. (A-G)** Dose-response curves depicting the percent proliferation of WT (black) or atovaquone-resistant (ATV[R], red) *T. gondii* parasites in the presence of increasing concentrations of **(A)** atovaquone, **(B)** buparvaquone, **(C)** auranofin, **(D)** trifloxystrobin, **(E)** azoxystrobin, **(F)** MMV024397, or **(G)** MMV688853. Values are expressed as a percent of the average fluorescence from a no-drug control at mid-log phase growth in the fluorescence proliferation assay, and represent the mean ± SEM of three (or four for (E)) independent experiments performed in triplicate; error bars that are not visible are smaller than the symbol. Inset bar graphs depict the $EC_{50}$ ± SEM (nM) of three (or four for (E)) independent experiments. Paired t-tests were performed and *p*-values are shown.

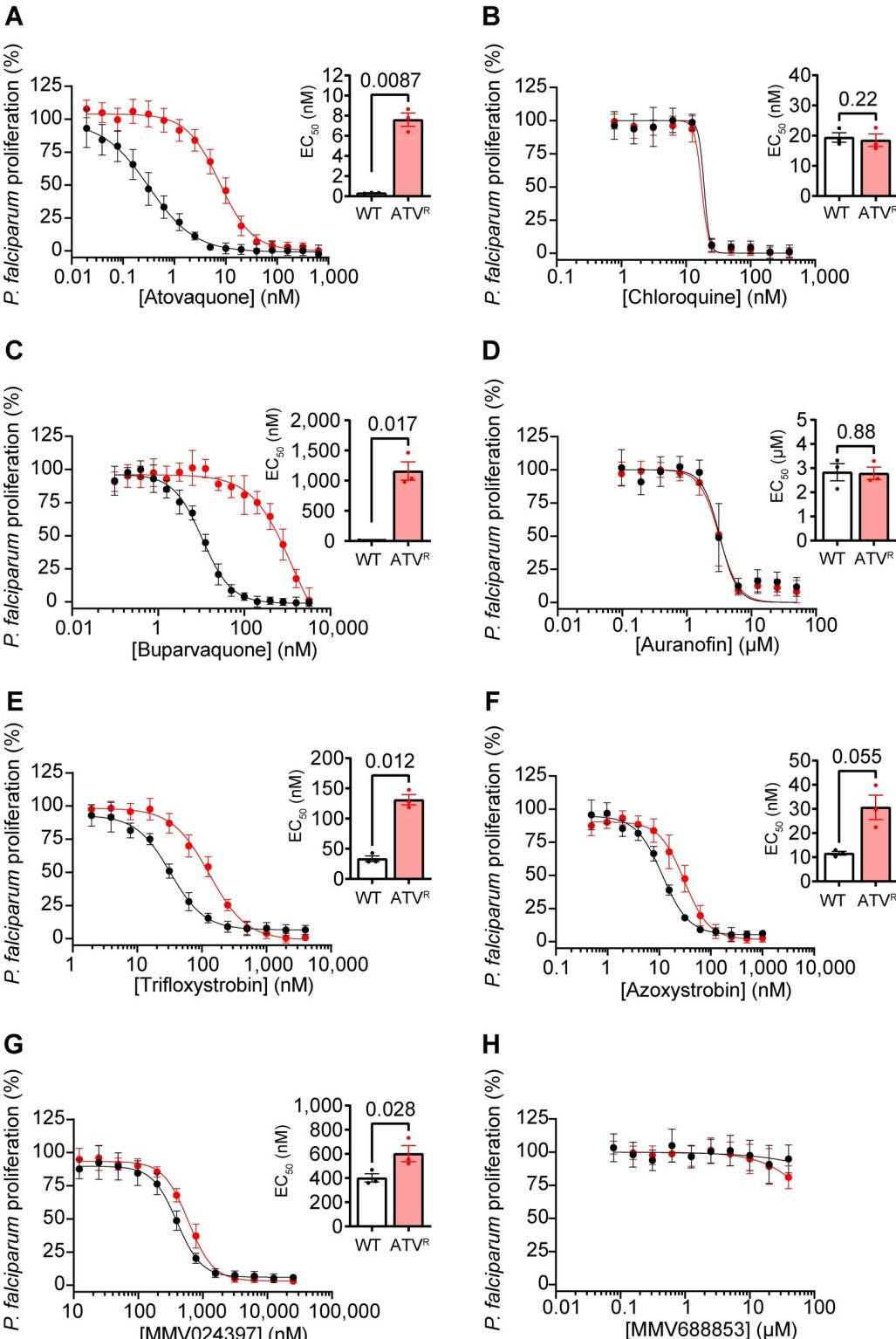

**Fig 8. Assessing the activity of ETC inhibitors against atovaquone-resistant *P. falciparum* parasites. (A-H)** Dose-response curves depicting the percent proliferation of WT (black) or atovaquone-resistant (ATV^R, red) *P. falciparum* parasites in the presence of increasing concentrations of **(A)** atovaquone, **(B)** chloroquine, **(C)** buparvaquone, **(D)** auranofin, **(E)** trifloxystrobin, **(F)** azoxystrobin, **(G)** MMV024397, or **(H)** MMV688853 after 96 h of culture, as measured using a SYBR Safe-based proliferation assay. Values are expressed as a percent of the average fluorescence from the no-

drug control, and represent the mean ± SEM of three independent experiments performed in triplicate; error bars that are not visible are smaller than the symbol. Inset bar graphs depict the $EC_{50}$ ± SEM of three independent experiments. Paired t-tests were performed and $p$-values are shown.

parental strain. We then performed fluorescence proliferation assays to compare the ability of ELQ-300 and the MMV compounds to inhibit parasite proliferation.

As expected, the $ELQ^R$ strain exhibited a ~7-fold higher $EC_{50}$ value for ELQ-300 compared to the $ELQ^S$ strain, similar to the extent of resistance reported previously (Table 1; Fig 9A; [45]). $ELQ^R$ parasites also exhibited ~6-fold resistance to the known $Q_i$ site inhibitor antimycin A (Table 1; Fig 9B). $ELQ^R$ parasites were not cross-resistant to any of the compounds identified in our screen, with the exception of MMV688853, which showed a small but significant two-fold increase in $EC_{50}$ ($p$ = 0.031; Table 1, Fig 9I). Curiously, compared to the $ELQ^S$ strain, $ELQ^R$ parasites were sensitized to many of the other inhibitors that we tested (Table 1, Fig 9). For example, $ELQ^R$ parasites exhibited significant ~5-fold greater sensitivity to atovaquone than the $ELQ^S$ strain ($p$ = 0.0025; Table 1; Fig 9C). Together, these data indicate that $Q_i$ site mutants that are resistant to $Q_i$ site-targeting compounds like ELQ-300 and antimycin A remain sensitive to most of the inhibitors we identified in our screen.

### MMV688853 likely targets the $Q_i$ site of Complex III in *T. gondii*

Our previous data indicate that the aminopyrazole carboxamide MMV688853 targets Complex III of the ETC (Fig 5). Given that MMV688853 also targets *Tg*CDPK1 ([33,34] and Fig 4), the observation that $ATV^R$ and $ELQ^R$ parasites exhibit minimal cross-resistance to MMV688853 in proliferation assays (Table 1; Figs 7G and 9I) could be explained by its ability to additionally target *Tg*CDPK1. Our data therefore leave open the possibility that resistance conferring mutations in the $Q_o$ or $Q_i$ sites of Complex III confer cross-resistance to the ETC-inhibiting properties of MMV688853. We tested whether $ATV^R$ or $ELQ^R$ parasites exhibited cross-resistance to MMV688853 in OCR assays. We first measured OCR in WT and $ATV^R$ parasites at a range of atovaquone, ELQ-300 and MMV688853 concentrations. As expected, $ATV^R$ parasites exhibited a significant, 52-fold increase in the $EC_{50}^{OCR}$ against atovaquone ($p$ = 0.041; Table 4; Fig 10A) whereas sensitivity to ELQ-300 was unchanged (Table 4; Fig 10B). We observed minimal differences in $EC_{50}^{OCR}$ of MMV688853 between WT and $ATV^R$ parasites (Table 4; Fig 10C). These data indicate that atovaquone-resistant Complex III, which contains an I254L mutation in the $Q_o$ site of the complex, remains sensitive to MMV688853.

Next, we measured OCR in $ELQ^S$ and $ELQ^R$ parasites at a range of atovaquone, ELQ-300 and MMV688853 concentrations. As expected, $ELQ^R$ parasites were highly resistant to OCR inhibition by ELQ-300 (Table 4; Fig 10E). Compared to $ELQ^S$ parasites, we observed a small but significant increase in sensitivity of $ELQ^R$ parasites to atovaquone ($p$ = 0.013; Table 4; Fig 10D), mirroring the slightly increased sensitivity of $ELQ^R$ parasites to atovaquone in the parasite proliferation experiments (Fig 9C). Notably, $ELQ^R$ parasites were highly resistant to OCR inhibition by MMV688853 (Table 4, Fig 10F). These data indicate that a $Q_i$ site mutation that confers resistance to ELQ-300 also confers resistance to MMV688853, consistent with the hypothesis that MMV688853 targets the $Q_i$ site of Complex III.

## Discussion

In this study, we screened the MMV 'Pathogen Box' compound library to identify inhibitors of the *T. gondii* ETC using a Seahorse XFe96 flux analyzer (Fig 1). One key benefit of using the Seahorse XFe96 flux analyzer as a drug-screening platform is that it simultaneously measures the $O_2$ consumption rate (OCR) and extracellular acidification rate (ECAR) of parasites to

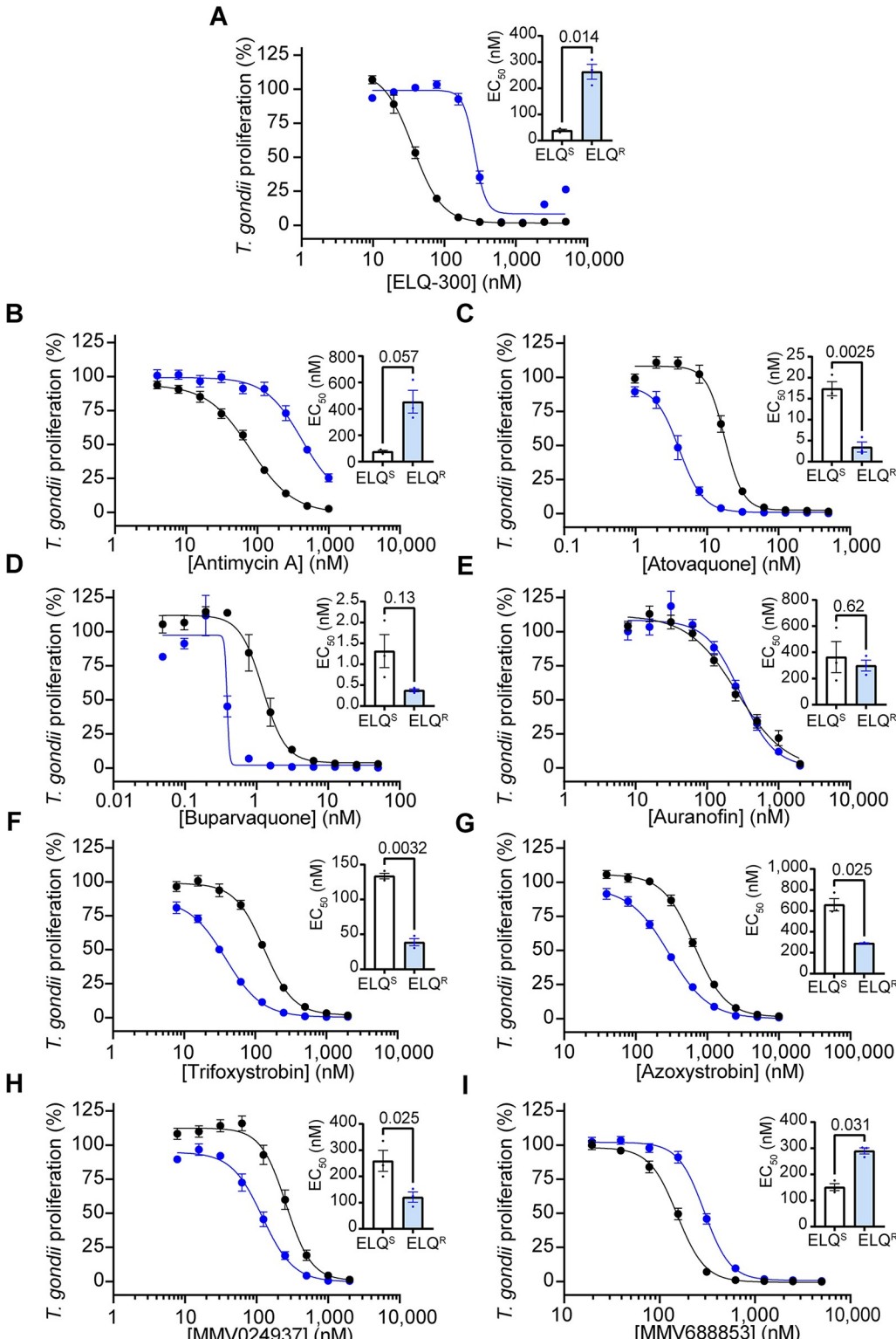

**Fig 9. Assessing the activity of ETC inhibitors against ELQ-300-resistant *T. gondii* parasites. (A-I)** Dose-response curves depicting the percent proliferation of ELQ-300-resistant (ELQ^R, blue) *T. gondii* parasites, or the corresponding ELQ-300-sensitive parental strain (ELQ^S, black), in the presence of increasing concentrations of **(A)** ELQ-300, **(B)** antimycin A, **(C)** atovaquone, **(D)** buparvaquone, **(E)** auranofin, **(F)** trifloxystrobin, **(G)** azoxystrobin, **(H)** MMV024397, or **(I)** MMV688853. Values are expressed as a percent of the average fluorescence from a no-drug control at mid-log

phase growth in the fluorescence proliferation assay, and represent the mean ± SEM of three independent experiments performed in triplicate; error bars that are not visible are smaller than the symbol. Inset bar graphs depict the $EC_{50} \pm$ SEM (nM) of three independent experiments. Paired t-tests were performed and *p*-values are shown.

assess ETC function and general metabolism, respectively. This enables on-target ETC inhibitors (*i.e.* those that inhibit OCR but not ECAR) to be differentiated from off-target compounds wherein the defect in OCR is a secondary effect resulting from rapid parasite death or otherwise impaired parasite metabolism (*i.e.* those that inhibit both OCR and ECAR). This is exemplified by the compound auranofin, which inhibited both OCR and ECAR of *T. gondii* and was subsequently shown to induce rapid parasite death (Fig 3). Furthermore, auranofin inhibited the proliferation of WT and yDHODH-expressing *P. falciparum* to a similar extent, indicating that its main target is not the ETC in these parasites (Fig 2). Although we cannot entirely rule out the possibility that auranofin has a direct effect on the ETC, taken together our data suggest that auranofin kills apicomplexan parasites via an ETC-independent process. Auranofin has been recently linked to the production of reactive oxygen species (ROS) in *T. gondii* [37]. Mitochondrial ROS can lead to impairment of ETC function in other organisms [46], which could explain the effects of auranofin on the ETC of *T. gondii*.

Another benefit of the screening approach that we have established is its scalability. By injecting three test compounds sequentially into each well, we were able to screen the entire 400 compound MMV 'Pathogen Box' using two 96-well Seahorse XFe96 plates. We note that it is possible to screen much larger compound libraries using this approach. A limitation of the screen is that we inject multiple compounds into the same well, which may mask inhibitors that are injected after 'hit' compounds, or lead to additive or confounding effects in compounds that we identify as hits. Follow-up tests of compounds injected following hit compounds can determine whether these too inhibit OCR, and secondary screens that test hits in isolation are important to further validate those compounds.

In addition to compound identification, our approaches enable a determination of where in the ETC identified inhibitors target. Using an assay to pinpoint the location of ETC defects in *T. gondii* [28,29], we demonstrated that most compounds identified in our screen (with the exception of auranofin) target ETC Complex III (Fig 5). Specifically, we demonstrated that: 1) the identified compounds inhibited OCR regardless of the electron source (malate or glycerol 3-phosphate) that was donating electrons to CoQ, implying that the inhibition occurred downstream of CoQ; and 2) a substrate that donates electrons directly to CytC (TMPD), and thereby bypasses Complex III, restored OCR, implying that the inhibition occurred upstream of CytC. The druggability of ETC Complex III in apicomplexan parasites has been noted before [13]. For instance, all seven novel hits identified in a screen for *Plasmodium* ETC inhibitors were found to target Complex III [26]. Our data do not rule out the possibility that, in addition to

**Table 4. $O_2$ consumption rate inhibitory activity of MMV688853 against drug-resistant *T. gondii* strains.** Determination of the $O_2$ consumption rate (OCR) inhibitory properties of MMV688853 against *T. gondii* parasite strains resistant to atovaquone (ATV^R ME49) or ELQ-300 (ELQ^R RH) and the corresponding ATV-sensitive (WT ME49) and ELQ-300-sensitive (ELQ^S) parental strains. Data are reported as average $EC_{50}^{OCR}$ (μM) ± SEM from three independent experiments. $EC_{50}^{OCR}$ values against atovaquone and ELQ-300 were determined as controls in each strain. FC, fold-change in $EC_{50}^{OCR}$ values between the atovaquone-resistant strain and corresponding atovaquone-sensitive parental strain. Paired t-tests were performed to compare the $EC_{50}^{OCR}$ of the compounds in WT vs ATV^R parasites, and *p*-values are depicted as ns = not significant ($p > 0.05$), * $p < 0.05$.

| | WT ME49 | ATV^R ME49 | | ELQ^S RH | ELQ^R RH |
|---|---|---|---|---|---|
| **Compound** | **$EC_{50}^{OCR}$ (μM)** | **$EC_{50}^{OCR}$ (μM)** | **FC** | **$EC_{50}^{OCR}$ (μM)** | **$EC_{50}^{OCR}$ (μM)** |
| MMV688853 | 1.9 ± 0.1 | 1.5 ± 0.1 | 0.8 * | 4.0 ± 0.7 | > 40 |
| Atovaquone | 0.027 ± 0.006 | 1.4 ± 0.3 | 52 * | 0.15 ± 0.02 | 0.09 ± 0.02 |
| ELQ-300 | 0.60 ± 0.1 | 0.52 ± 0.1 | 0.9 ns | 2.8 ± 0.7 | > 4 |

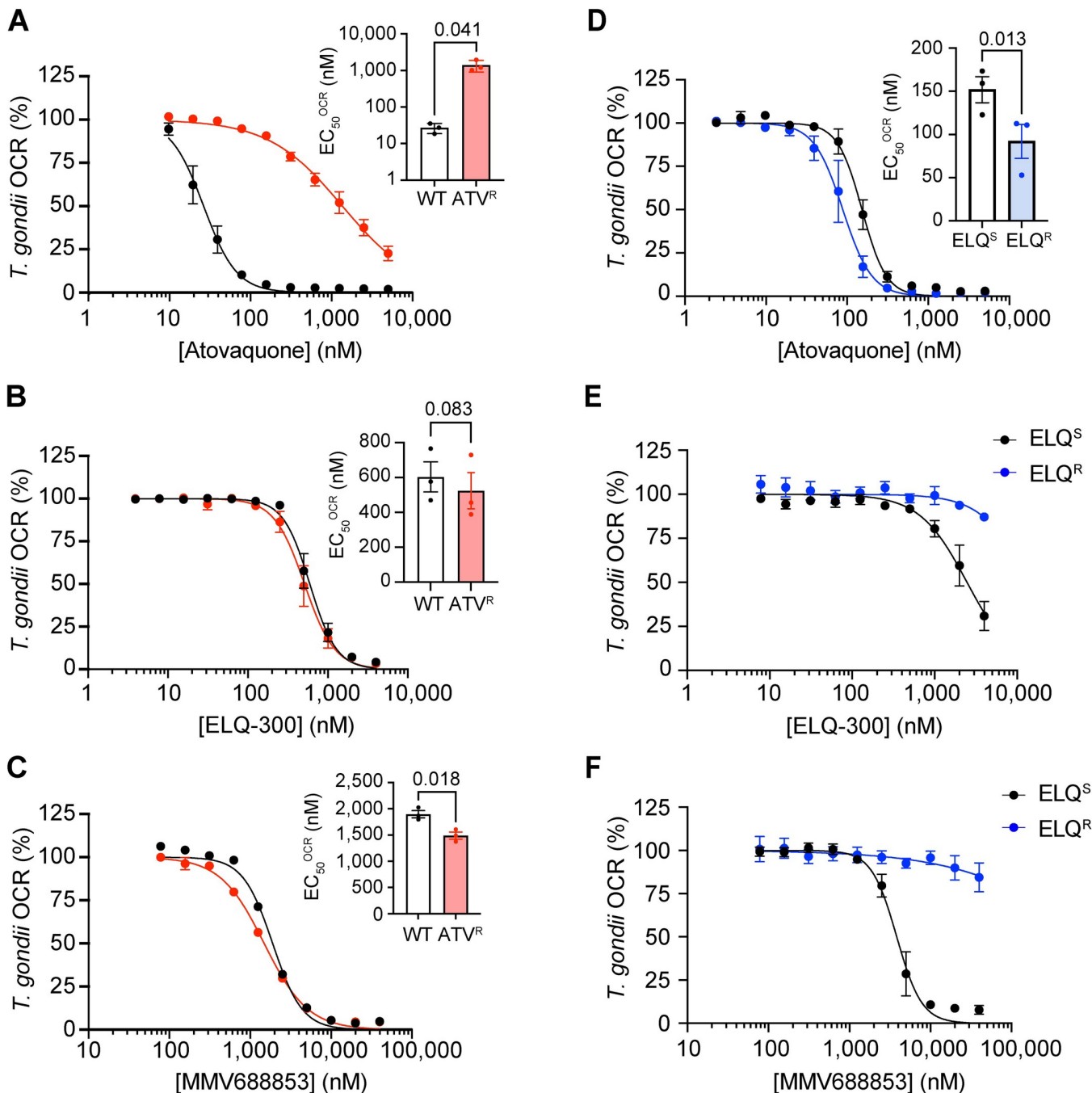

**Fig 10. ELQ-300-resistant parasites exhibit cross-resistance to MMV688853 in O$_2$ consumption rate activity assays. (A-C)** Dose-response curves depicting the OCR of intact WT (black) or atovaquone-resistant (ATV[R], red) *T. gondii* parasites in the presence of increasing concentrations of **(A)** atovaquone, **(B)** ELQ-300 or **(C)** MMV688853. **(D-F)** Dose-response curves depicting the OCR of intact parental ELQ-300 sensitive (ELQ[S]; black) or ELQ-300-resistant (ELQ[R], blue) *T. gondii* parasites in the presence of increasing concentrations of **(D)** atovaquone, **(E)** ELQ-300 or **(F)** MMV688853. Values represent the percent OCR relative to the no-drug (100% OCR) and inhibitory atovaquone-treated (D-F) or antimycin A-treated (A-D; 0% OCR) controls, and depict the mean ± SEM of three independent experiments, each conducted in at least duplicate; error bars that are not visible are smaller than the symbol. Inset bar graphs depict the EC$_{50}$[OCR] ± SEM (nM) of three independent experiments. Where relevant, paired t-tests were performed and *p*-values are shown.

inhibition of Complex III, the identified compounds also inhibit targets upstream in the ETC (*e.g.* one or more of the dehydrogenases that donate electrons to coenzyme Q). For instance, while the ETC inhibitor 1-hydroxy-2-dodecyl-4(*1H*)quinolone can target Complex III, it can also inhibit DHODH and the single subunit NADH dehydrogenases of apicomplexan parasites [47–50], likely by targeting the CoQ binding sites of each.

*P. falciparum* rapidly develops resistance to the Complex III inhibitor atovaquone when used in a clinical setting [41,42], and although atovaquone-resistant clinical isolates of *T. gondii* have not been characterized, patients treated with atovaquone frequently experience reactivation of toxoplasmosis [51–53]. Atovaquone resistance arises from mutations in the $Q_o$ site of the cytochrome *b* protein of Complex III [14,19,20,43]. We tested our identified inhibitors against atovaquone-resistant strains of both *T. gondii* and *P. falciparum* (Fig 7 and 8). We found that $ATV^R$ parasites exhibited extensive cross-resistance to buparvaquone, a structural analog of atovaquone [54], in both *T. gondii* (~223-fold; Fig 7B) and *P. falciparum* (~106-fold; Fig 8C). Notably, we found minimal cross-resistance to the other tested compounds (Table 1). For example, $ATV^R$ *P. falciparum* parasites have only mild cross-resistance, and $ATV^R$ *T. gondii* parasites have slightly increased sensitivity, to the strobilurin compounds trifloxystrobin and azoxystrobin (Figs 7D–7E and 8E-F; Table 1). Strobilurins have been shown to target the $Q_o$ site of Complex III in fungi [30], and a study that introduced *P. falciparum* $Q_o$ site residues into the yeast $Q_o$ site indicated that azoxystrobin may also target this site in apicomplexans [32]. Given the small shifts in $EC_{50}$ observed in the $Q_o$ site mutant in our study, our data suggest that if the strobilurins bind the $Q_o$ site, they may do so in a different manner to atovaquone and buparvaquone.

We additionally tested the identified ETC inhibitors against a strain of *T. gondii* containing a $Q_i$ site mutation in the cytochrome *b* protein of Complex III that confers resistance against a range of ELQ family inhibitors ($ELQ^R$; [44,45]). Like with $ATV^R$ parasites, $ELQ^R$ parasites did not exhibit cross-resistance, and in several instances exhibited increased sensitivity, to most of the inhibitors from our screen (Tables 1 and 4; Fig 9). Similar, small increases in sensitivity to other Complex III inhibitors have been observed in $ATV^R$ parasites from *P. falciparum* in a previous study [55]. Taken together, these data suggest that the chemically diverse compounds that we identified in our screen maybe useful in the treatment of $ATV^R$ and $ELQ^R$ parasitic infections. However, we note that several other $Q_o$ and $Q_i$ site mutations can confer atovaquone or ELQ resistance [12,19,20,56], and as such further, more comprehensive studies should test whether these compounds are effective against other $ATV^R$ and $ELQ^R$ strains.

Our screen identified two compounds that, to our knowledge, have not been characterized as ETC inhibitors before. The first of these is MMV024397 (6-(4-Benzylpiperidin-1-yl)-*N*-cyclopropylpyridine-3-carboxamide), a compound that is listed under the 'malaria' disease set of the MMV 'Pathogen Box' and shown to inhibit the proliferation of *P. falciparum* (Fig 2G) [35], but for which very little other information exists. We demonstrated that MMV024397 inhibited ETC function in both *T. gondii* and *P. falciparum* in a manner consistent with Complex III inhibition. Future studies exploring exactly how this compound inhibits Complex III are warranted.

The second novel ETC inhibiting compound we identified is the aminopyrazole carboxamide compound MMV688853, which has been characterized previously as an inhibitor of *Tg*CDPK1 [33,34]. Huang *et al.* (2015) generated a parasite strain in which the 'gatekeeper' residue of *Tg*CDPK1 was mutated (*Tg*CDPK1$^{G128M}$) to render *Tg*CDPK1 resistant to aminopyrazole carboxamides. They found that, despite this mutation, parasite proliferation could still be impaired by several aminopyrazole carboxamide derivatives of MMV688853, suggesting an additional target for these compounds. Our data reveal that MMV688853 targets the $Q_i$ site of Complex III of the ETC (Figs 4, 5 and 10). Given that we observe no significant shift in the

EC$_{50}$ of MMV688853 in parasites where $Tg$CDPK1 has been engineered to be resistant to this compound (Fig 4F), our data suggest that Complex III is a major target of MMV688853, and potentially other aminopyrazole carboxamides, in the parasite. Mutations in cytochrome $b$ can lead to the rapid emergence of resistance to Complex III inhibitors such as atovaquone [19], and it will be of interest to explore whether the dual-targeting properties of MMV688853 make $T$. $gondii$ less prone to developing resistance. It will also be of interest to screen other aminopyrazole carboxamide compounds and/or perform structure-activity relationship studies to determine the chemical basis for the dual inhibition of $Tg$CDPK1 and Complex III by MMV688853.

We found that MMV688853 failed to inhibit proliferation (Table 2; Fig 2H) or O$_2$ consumption (Table 3; Figs 6G and S9F) of $P$. $falciparum$ at the concentration ranges we tested (up to 40 μM for proliferation and 50 μM for O$_2$ consumption). The difference in activity of this compound against $T$. $gondii$ and $P$. $falciparum$ is curious. It is conceivable that these differences are due to impaired uptake of MMV688853 into $P$. $falciparum$ parasites. However, given that we performed the OCR assays with plasma membrane-permeabilized $P$. $falciparum$ parasites, this explanation is unlikely. A previous study found that MMV688853 was particularly potent against the ookinete stage of $P$. $berghei$ (EC$_{50}$ 220 nM) [57]. The ookinete is the motile zygote that forms in the midgut of the mosquito vector shortly after transmission of the parasite from the vertebrate host. The potency of MMV688853 against ookinetes was suggested to result from its targeting the $Plasmodium$ homolog of $Tg$CDPK1, which is proposed to play a key role in transmission of the parasite into the insect stages of the life cycle [58]. However, given its dual activity, it is also conceivable that MMV688853 targets the ETC of $Plasmodium$, which becomes more active in the insect stages of the parasite life cycle [50,59]. At odds with this hypothesis is that Complex III is essential in both insect and vertebrate life stages of $Plasmodium$ [9,50,59]. A final possibility is that there are structural differences in the Q$_i$ site of Complex III between $P$. $falciparum$ and $T$. $gondii$ that results in the inability of MMV688853 to target the Q$_i$ site in $P$. $falciparum$. Of note, we found that a single Q$_i$-site mutation in $T$. $gondii$ can render Complex III resistant to MMV688853, suggesting the Complex III-targeting properties of this compound are susceptible to small changes in Q$_i$ site structure. Understanding these differences will be a priority for future research.

In summary, our work has developed a scalable pipeline to screen compound libraries to identify inhibitors of the ETC in apicomplexan parasites and characterize their targets. We identified chemically diverse Complex III inhibitors, including MMV688853, which our data suggest is a dual Complex III and $Tg$CDPK1 inhibitor. As many of the identified Complex III inhibitors were active against atovaquone-resistant $T$. $gondii$ and $P$. $falciparum$, these findings will aid in the development of much-needed new therapeutics against these parasites.

## Materials and methods

### Host cell and parasite culture, and genetic manipulation

Tachyzoite-stage $T$. $gondii$ parasites were cultured in human foreskin fibroblasts (HFF) in Dulbecco's modified Eagle's medium (DMEM) containing 2 g/L NaHCO$_3$, supplemented with 1% (v/v) fetal calf serum, 50 units/mL penicillin, 50 μg/mL streptomycin, 10 μg/mL gentamicin, 0.25 μg/mL amphotericin B, and 0.2 mM L-glutamine. RH strain $T$. $gondii$ parasites expressing the tandem dimeric Tomato (tdTomato) red fluorescent protein [60] were used in the initial drug screening assays and for most subsequent $T$. $gondii$ experiments. For the atovaquone resistance experiments, we used wild type ME49 strain parasites or atovaquone-resistant ME49 strain parasites (clone R32), both described previously ([19]; a kind gift from Michael Panas and John Boothroyd, Stanford University). For the ELQ-300 resistance experiments, we

used a RH parasite strain modified to express GFP and β-galactosidase and selected for resistance to the ELQ-300 analog ELQ-316 (which we termed ELQ$^R$), and the corresponding ELQ-sensitive, GFP- and β-galactosidase-expressing parental strain (ELQ$^S$), both described previously ([45]; a kind gift from Stone Doggett, Oregon Health and Science University). We re-derived the ELQ$^R$ strain after noticing that the existing ELQ$^R$ parasite population we were culturing consisted of both ELQ-300 resistant and ELQ-300 sensitive parasites. We selected parasites for ~3 weeks on 50 nM ELQ-300, then re-cloned parasites. To verify that the resulting parasites contained the expected ELQ-300 resistant or sensitive allele, we sequenced the cytochrome $b$ gene from cDNA synthesized using random hexamers from total RNA extracted from both the ELQ$^R$ and ELQ$^S$ strains as per the manufacturer's instructions (Superscript IV first-strand synthesis system, Thermo Fisher Scientific). We amplified cytochrome $b$ from the ELQ$^R$ and ELQ$^S$ strains using the oligonucleotides 5'-ATGAGTCTATTCCGGGCACA and 5'-GTATAAGCATAGAACCAATCCGGT and sequenced the resulting product by Sanger sequencing using the same oligonucleotides. To allow us to undertake fluorescence proliferation assays with the ME49 WT, ATV$^R$, ELQ$^S$ and ELQ$^R$ parasites, we introduced a tdTomato-encoding vector into these strains using fluorescence-activated cell sorting, as described previously [39].

To introduce a glycine to methionine mutation at residue 128 of the $Tg$CDPK1 protein of $T. gondii$ parasites ($Tg$CDPK1$^{G128M}$), we used a CRISPR-Cas9-based genome editing strategy. We introduced a single guide RNA (sgRNA) targeting the desired region of the open reading frame of the $tg$cdpk1 gene into the pSAG1::Cas9-U6-UPRT vector (Addgene plasmid 54467; [61]) using Q5-site directed mutagenesis according to the manufacturer's instructions (New England Biolabs). We performed the Q5 reaction using the following primers 5'-<u>AAAGGCTA CTTCTACCTCGT</u>GTTTTAGAGCTAGAAATAGCAAG-3' (sgRNA coding region underlined) and 5'-AACTTGACATCCCCATTTAC-3'. We also generated a double stranded donor DNA encoding the $Tg$CDPK1$^{G128M}$ mutation flanked by 42–45 bp of homologous flanks to either side of the target site. To do this, we annealed the oligonucleotides 5'-CTGTATGAAT TCTTCGAGGACAAAGGCTACTTCTACCTCGTCatgGAAGTGTACACGGGAGGCGAG TTGTTCGACGAGATCATTTCCCGC-3' and 5'-GCGGGAAATGATCTCGTCGAACAAC TCGCCTCCCGTGTACACTTCcatGACGAGGTAGAAGTAGCCTTTGTCCTCGAAGAA TTCATACAG-3' (mutated base pairs are indicated by the lower case letters). We combined the sgRNA expressing plasmid (which also encodes Cas9-GFP) and donor DNA and transfected them into TATiΔ$ku80$/Tomato$^+$ parasites by electroporation as described previously [62]. Two days after transfection, we selected and cloned GFP$^+$ parasites by flow cytometry. We PCR-amplified the genomic DNA of several clones using the primers 5'-AGTGAAGCAGAAGACG GACAAG-3' and 5'-GAGGTCCCGATGTACGATTTTA-3', and checked for successful modification by Sanger sequencing. We termed the resulting parasite strain '$Tg$CDPK1$^{G128M}$'.

Asexual blood stages of 3D7 strain $P. falciparum$ parasites were maintained in synchronous continuous culture using O$^+$ human erythrocytes in Roswell Park Memorial Institute (RPMI)-1640 medium supplemented with 25 mM HEPES, 20 mM D-glucose, 200 μM hypoxanthine, 24 mg/L gentamicin and Albumax II (0.6% w/v), as described previously [63,64]. Atovaquone-resistant parasites were generated by maintaining cultures at 1% parasitaemia in the presence of atovaquone at an initial concentration equivalent to the EC$_{50}$ of atovaquone (0.5 nM). Fresh medium, erythrocytes and atovaquone were added every 2 days and parasitaemia was adjusted to 1%. The atovaquone concentration was increased gradually across 12 weeks. Once parasites were proliferating in the presence of 10 nM atovaquone (~20× EC$_{50}$), clonal populations were selected by limiting dilution cloning. We PCR-amplified the cytochrome $b$ gene of $P. falciparum$ using primers described previously [65]: 5'-CTCTATTAATTTAGTTAAAGCACAC-3' and 5'-ACAGAATAATCTCTAGCACC-3'. We checked for mutations in the amplified

cytochrome *b* gene by Sanger sequencing using the following primers: 5′-AGCAGTAATTTG-GATATGTGGAGG-3′ and 5′-AATTTTTAATGCTGTATCATACCCT-3′. 3D7 strain *P. falciparum* parasites expressing yeast dihydroorotate dehydrogenase (yDHODH) were a kind gift from Emily Crisafulli and Stuart Ralph (University of Melbourne), and were maintained on 10 nM WR99210 (which was removed prior to growth assays) as described previously [66].

## Compounds

The 'Pathogen Box' compounds were kindly provided by MMV in 96-well plates containing 10 mM stock solutions dissolved in dimethyl sulfoxide (DMSO). Additional amounts of several compounds were purchased from Sigma Aldrich and dissolved in DMSO (stock concentration given in brackets), including azoxystrobin (31697-100MG; 50 mM), trifloxystrobin (46447-100MG; 50 mM), auranofin (A6733-10MG; 50 mM), buparvaquone (SML1662-25MG; 3 mM), and atovaquone (A7986-10MG; 10 mM). 3-MB-PP1 was purchased from Cayman Chemical (17860; 10 mM). Additional MMV688853 (BKI-1517; 10 mM) was a kind gift from Wes Van Voorhis (University of Washington). Additional MMV024397 was also provided by MMV. The DMSO concentration introduced when using these compounds in assays was < 0.2% (v/v), except MMV688853 when used at the higher concentrations (up to 50 μM) in the *Plasmodium* assays (up to 0.5% (v/v) DMSO).

## Screening compounds using a Seahorse XFe96 extracellular flux assay

The MMV 'Pathogen Box' compounds were screened for their ability to inhibit $O_2$ consumption of intact *T. gondii* parasites using a Seahorse XFe96 flux assay described previously [28] with slight modifications. Parasites (tdTomato-expressing RH strain *T. gondii* tachyzoites) were mechanically egressed from host cells by passing them through a 26-gauge needle, then filtered through a 3 μm polycarbonate filter to remove host cell debris, counted using a hemocytometer, and pelleted by centrifugation ($1,500 \times g$, 10 min, RT). The medium was aspirated and parasites were washed once in Base Medium (Agilent) supplemented with 1 mM L-glutamine and 5 mM D-glucose (termed 'supplemented Base Medium'), then resuspended in supplemented Base Medium to $1.5 \times 10^7$ parasites/mL. Parasites ($1.5 \times 10^6$) were seeded into wells of a Seahorse XFe96 cell culture plate coated with 3.5 μg/cm$^2$ CellTak cell adhesive (Corning) and attached to the bottom by centrifugation ($800 \times g$ for 3 min). The final well volume was 175 μL, achieved by adding supplemented Base Medium. MMV 'Pathogen Box' compounds were prepared such that the final concentration upon injection (25 μL injection volumes) would be 1 μM (8 μM for compounds to be injected from port A; 9 μM for compounds to be injected from port B; and 10 μM for compounds to be injected from port C). During the XFe96 assay, three cycles of 1 min mixing and 3 min measuring were taken to determine the baseline OCR. Three compounds were sequentially injected into each well (from ports A-C) and the OCR measured for four cycles of 1 min mixing followed by 3 min measuring. A final injection of the known ETC Complex III inhibitors antimycin A (10 μM) and atovaquone (1 μM) from port D was used as a control to validate that the assay was measuring mitochondrial OCR, and to enable determination of non-mitochondrial OCR. In some instances where 'hit' compounds were injected from ports A or B, compounds injected from later ports in that particular well were retested in a subsequent assay to ensure compounds injected after the 'hit' compound were not missed. Percent inhibition of OCR by each of the tested compounds was calculated relative to the antimycin A- and atovaquone-treated control (100% inhibition) and the OCR measurement taken prior to compound injection (0% inhibition). An arbitrary cut-off of >30% inhibition of OCR was applied in selecting candidate ETC inhibitors from the screen. All raw and calculated data from the screen are presented in S1 Table. To determine a

confidence value for the separation between 'hit' and 'non-hit' compounds in our screen, we calculated the Z'-factor for the assay [67]. We found this to be 0.3 for each plate. This suggests a small separation between hit and non-hit compounds, necessitating subsequent validation assays for each candidate hit compound identified.

## Seahorse XFe96 extracellular flux analysis of intact *T. gondii* parasites

The inhibitory activity of selected MMV 'Pathogen Box' compounds against the OCR of intact *T. gondii* parasites was assessed using a previously described Seahorse XFe96 flux assay [28] with slight modifications. *T. gondii* parasites were prepared and seeded into wells of a Cell-Tak coated Seahorse XFe96 cell culture plate as described above. The final well volume was 175 μL, achieved with supplemented Base Medium. Carbonyl cyanide 4-(trifluoromethoxy)phenylhydrazone (FCCP) was prepared in Base Medium such that the final concentration upon injection would be 1 μM (8 μM for injection from port A). A serial dilution of the test compounds as well as a no-drug (DMSO) control was performed in supplemented Base Medium, and loaded into port B at 9× the desired final concentrations. Supplemented Base Medium was injected from port C, and a final injection of the known ETC Complex III inhibitors atovaquone or antimycin A (5 μM or 20 μM final concentrations, respectively) from port D was used as a control to completely inhibit mitochondrial OCR. The OCR and ECAR were measured for three cycles of 30 s mixing followed by 3 min measuring at baseline after injections from port A and port C, and for six cycles of 30 s mixing followed by 3 min measuring after injections from port B and port D. Mitochondrial OCR was calculated by subtracting the last OCR reading after atovaquone injection (port D) from the last OCR reading after test compound injection (port B). Percent OCR relative to the drug-free control was plotted against the test compound concentration, and a sigmoidal four parameter logistic curve or a variable slope [inhibitor] vs normalized response curve was fitted to the data using nonlinear regression in GraphPad Prism to estimate the effective compound concentration required for 50% inhibition of *T. gondii* OCR ($EC_{50}^{OCR}$).

## Seahorse XFe96 extracellular flux analysis of plasma membrane-permeabilized parasites

Measurement of substrate-elicited OCR of digitonin-permeabilized *T. gondii* parasites was performed as described previously [28,29]. Briefly, freshly egressed *T. gondii* parasites were passed through a 3 μm filter to remove host cell debris, counted using a hemocytometer, and pelleted by centrifugation (1,500 × g, 10 min, RT). Parasites were washed once in non-supplemented Base Medium, resuspended in non-supplemented Base Medium to $1.5 \times 10^7$ parasites/ mL and incubated at 37°C for approximately 1 hour to deplete endogenous substrates. Parasites were added to the wells of a Cell-Tak-coated Seahorse cell culture plate at a density of $1.5 \times 10^6$ parasite per well and centrifuged (800 × g, 3 min, RT) to adhere parasites to the bottom of the wells. Just before the beginning of the assay, Base Medium was removed and replaced with 175 μL mitochondrial assay solution (MAS) buffer (220 mM mannitol, 70 mM sucrose, 10 mM $KH_2PO_4$, 5 mM $MgCl_2$, 0.2% (w/v) fatty acid-free bovine serum albumin (BSA), 1 mM EGTA and 2 mM HEPES-KOH pH 7.4) containing 0.002% (w/v) digitonin to permeabilize the parasite plasma membrane. The following compounds were prepared in MAS buffer (final concentration after injection given in brackets) and loaded into ports A-D of the XFe96 sensor cartridge: Port A, ETC substrates malate (Mal; 10 mM) or sn-glycerol 3-phosphate bis(cyclohexylammonium) salt (G3P; 25 mM) plus FCCP (1 μM); Port B, the test compounds atovaquone (1.25 μM), auranofin (10 μM), azoxystrobin (20 μM), trifloxystrobin (2.5 μM), MMV688853 (20 μM), buparvaquone (5 μM) or MMV024397 (20 μM); Port C, *N*,*N*,

$N'$,$N'$-tetramethyl-$p$-phenylenediamine dihydrochloride (TMPD; 0.2 mM) mixed with ascorbic acid (3.3 mM); Port D, sodium azide (NaN$_3$; 10 mM). The OCR was assessed for three cycles of 30 s mixing followed by 3 min measuring to establish baseline OCR before substrate injection, for three cycles of 30 s mixing followed by 3 min measuring after the injection of substrates from ports A and C, and for six cycles of 30 s mixing followed by 3 min measuring after the injections of compounds from ports B and D. A minimum of four background wells (containing no parasites) were used in each plate, and 2 or 3 technical replicates were used for each condition.

OCR measurements of digitonin-permeabilized *P. falciparum* parasites were performed using a protocol modified from one described previously [68]. On the day of the assay, 200 mL of *P. falciparum* culture at 4% (v/v) hematocrit and at least 5% parasitemia was enriched for trophozoites by passing through a MACS CS column placed in the magnetic field of a Super-MACS II (Miltenyi Biotec) separator according to the manufacturer's instructions. The trophozoites were freed from erythrocytes by treating with 0.05% (w/v) saponin at 37°C for 5 minutes. The obtained parasite pellets were washed with phosphate buffered saline (PBS) until the supernatant was no longer red (*i.e.* until most host cell hemoglobin had been removed). Parasites were counted using a hemocytometer and prepared at $5 \times 10^7$ parasites/mL in MAS buffer supplemented with 10 mM malate and 0.002% (w/v) digitonin. Parasites were seeded at a density of $5 \times 10^6$ cells per well in a Cell-Tak-coated XFe96 cell culture plate and centrifuged ($800 \times g$, 3 min, RT) to adhere the parasites to the bottom of the wells. Supplemented MAS buffer (75 μL) was carefully added to the wells without disturbing the cell monolayer. The following compounds were prepared in MAS buffer (final concentration after injection given in brackets) and loaded into ports A-C of the XFe96 sensor cartridge: Port A, a 2-fold serial dilution of the test compounds; Port B, TMPD (0.2 mM) mixed with ascorbic acid (2 mM); Port C, sodium azide (NaN$_3$; 10 mM). The OCR was measured for five cycles of 20 s mixing, 1 min waiting, 2.5 min measuring at baseline and after each injection. Percent OCR relative to the drug-free control was plotted against the test compound concentration, and a variable slope (four parameters) curve was fitted using nonlinear regression in GraphPad Prism to yield the EC$_{50}$ for *P. falciparum* OCR (EC$_{50}^{OCR}$).

## *T. gondii* fluorescence proliferation assays

The anti-parasitic activity of selected MMV 'Pathogen Box' compounds was assessed by fluorescence proliferation assays, measuring the proliferation of tdTomato-expressing *T. gondii* parasites as described previously [39]. Briefly, 2,000 parasites were added to wells of a clear bottom, black 96-well plate containing HFF cells, in phenol red-free DMEM supplemented with 1% (v/v) fetal calf serum, 50 units/mL penicillin, 50 μg/mL streptomycin, 10 μg/mL gentamicin, 0.25 μg/mL amphotericin B, and 0.2 mM L-glutamine. A serial dilution of the test compounds at the desired concentrations was performed and added to wells of the plate. Parasites were allowed to proliferate and fluorescence was measured once or twice daily using a FLUOstar OPTIMA Microplate Reader (BMG LABTECH). Percent parasite proliferation relative to the no-drug control at mid-log phase was plotted against the compound concentration, and a variable slope (four parameters) curve was fitted using nonlinear regression in GraphPad Prism, enabling calculation of the EC$_{50}$ of the compound against *T. gondii* proliferation.

## *P. falciparum* proliferation assays

The anti-plasmodial activity of selected MMV 'Pathogen Box' compounds was assessed using a SYBR Safe-based fluorescence assay described previously [21,69]. Assays were set up using ring-stage *P. falciparum*-infected erythrocytes in culture medium at a hematocrit of 1% and

parasitemia of 0.5%. Parasites were allowed to proliferate for 96 h, after which the percentage parasite proliferation was plotted against the compound concentration. A variable slope (four parameters) curve was fitted to the data using nonlinear regression in GraphPad Prism, enabling calculation of the $EC_{50}$ of the compound against *P. falciparum* proliferation.

### Flow cytometry analysis of *T. gondii* viability

Freshly egressed RH$\Delta$*hxgprt* strain *T. gondii* parasites were passed through a 3 μm filter to remove host cell debris. Parasites were pelleted by centrifugation (1,500 × *g*, 10 min, RT) and resuspended in phenol red-free DMEM containing 5 mM D-glucose and 1 mM L-glutamine. Parasites were incubated (37˚C, 5% $CO_2$) for various times (15 to 120 min) in the presence of DMSO (vehicle control), auranofin (1 μM, 20 μM or 100 μM) or atovaquone (10 μM). Propidium iodide (PI, 15 μM) was then added and parasites were incubated for a further 20 min (RT, protected from light), before being analyzed on a BD LSR II Flow Cytometer. PI fluorescence was excited using the 488 nm laser and detected with a 670/14 nm filter. Acquired data were exported for further analysis using FlowJO 10 (BD) software. FSC and SSC parameters were used to gate for single parasites, with the gating strategy outlined in S5 Fig. Parasite viability was determined as the percentage of PI negative parasites in the total parasite population (PI negative and positive) normalized to the percentage of PI negative parasites in the DMSO treated control at each time point assayed.

### Human cell proliferation assay

The inhibitory activities of the selected MMV 'Pathogen Box' compounds against a primary human cell line were assessed using a confluence-based proliferation assay. Assays were set up using human foreskin fibroblast (HFF) cells in DMEM supplemented with 10% (v/v) newborn calf serum, 50 units/mL penicillin, 50 μg/mL streptomycin, 10 μg/mL gentamicin, 0.25 μg/mL amphotericin B, and 0.2 mM L-glutamine at a seeding density of approximately 5,000 cells/ well in a 96-well plate. A serial dilution of the desired compounds was performed and added to wells of the plate. Cells were allowed to proliferate for 90–96 hours in an IncuCyte FLR system (Essen Bioscience) housed in a humidified, 37˚C incubator at 5% $CO_2$ before the confluency of each well was assessed. The percentage cell proliferation relative to a no-drug control was plotted against the compound concentration. A variable slope (four parameters) curve was fitted to the data using nonlinear regression in GraphPad Prism, enabling calculation of the $EC_{50}$ of the compounds against HFF proliferation.

### Complex III enzymatic assay

To measure Complex III enzymatic activity in *T. gondii*, we adapted an assay previously established for mammalian cells [70]. Egressed parasites were passed through a 5 μm polycarbonate filter to remove host cell debris, counted using a hemocytometer, and pelleted by centrifugation (10 min, 1,500 × *g*, RT). Pellets were washed in 1 mL cold PBS and centrifuged (1 min, 12,000 × *g*, RT). Parasites were resuspended to $2.5 \times 10^8$ parasites/mL in MAS buffer containing 0.2% (w/v) digitonin, and lysed on a spinning wheel (30 min, 4˚C). Complex III assay buffer (25 mM $KH_2PO_4$ pH 7.5, 75 μM oxidised equine heart cytochrome *c*, 100 μM EDTA, 0.025% (v/v) Tween-20 and 1.21 mM sodium azide) was prepared and aliquoted into the wells of a 24-well plate. The following compounds (or DMSO as a no-drug vehicle control) were added to three wells each at the indicated final concentrations: atovaquone (1.25 μM), buparvaquone (5 μM), auranofin (10 μM), trifloxystrobin (2.5 μM), azoxystrobin (20 μM), MMV024397 (20 μM) and MMV688853 (20 μM).

A baseline reading was taken by measuring the absorbance at 550 nm every 15 s for 2 min using a TECAN Infinite 200 PRO plate reader warmed to 37˚C. Parasite lysate (an equivalent of $6.25 \times 10^6$ parasites per mL) was then added to two of the three wells per drug (duplicate technical experimental wells) while MAS buffer was added to the remaining well (as a 'no parasite lysate' background control), and a further baseline reading was taken every 15 s for 2 min. To start the reaction, 5 μM reduced decylubiquinol in DMSO was added to each well, and absorbance at 550 nm was measured every 15 s for 60 min.

To calculate enzymatic activity, absorbance was plotted as a function of time. The initial rate was estimated from the first 5 minutes after adding decylubiquinol, and divided by the extinction coefficient for reduced cytochrome $c$ (18.5 mM$^{-1}$ cm$^{-1}$) according to the Beer-Lambert Law. For each condition, the background (initial rate in the absence of parasite lysate) was subtracted from the observed value to yield the calculated activity.

### *T. gondii* invasion assay

To determine the effects of compounds on parasite invasion, we undertook invasion assays based on a modified version of a previously described protocol [71]. TATiΔ*ku80*/Tomato$^+$ or *Tg*CDPK1$^{G128M}$/Tomato$^+$ strain *T. gondii* parasites were cultured in HFF cells such that most parasites were still intracellular prior to the assay. Extracellular parasites were removed by washing the flask three times with warm intracellular (IC) buffer (5 mM NaCl, 142 mM KCl, 2 mM EGTA, 1 mM MgCl$_2$, 5.6 mM D-glucose and 25 mM HEPES, pH 7.4). Infected host cells were then scraped from the flasks, passed through a 26-gauge needle to mechanically egress the parasites, and filtered through a 3 μm polycarbonate filter to remove host cell debris. Parasites were counted using a hemocytometer and diluted to $5 \times 10^5$ parasites per mL in IC buffer with either DMSO (vehicle control), MMV688853 (5 μM), 3-MB-PP1 (5 μM) or atovaquone (1 μM), added to wells of a 24-well plate containing confluent HFF cells cultured on coverslips, and incubated at 37˚C for 45 min to allow parasites to attach to host cells. To induce invasion, IC buffer was removed and replaced with DMEM containing DMSO/drug added at the above concentrations. Parasites were allowed to invade for 25 min at 37˚C, before being fixed in 3% (w/v) paraformaldehyde (PFA) and 0.1% (w/v) glutaraldehyde in PBS for 20 min at RT. After fixation, coverslips were blocked in 2% (w/v) BSA in PBS. To identify uninvaded extracellular parasites, we conducted immunofluorescence assays. We labelled uninvaded extracellular parasites with the *T. gondii* cell surface marker mouse anti-SAG1 primary antibody (Abcam, Ab8313; 1:1,000 dilution) and a goat anti-mouse Alexa Fluor 488 Plus secondary antibody (Thermo Fisher Scientific, A32723; 1:500 dilution). Coverslips were mounted onto slides, the identity of the samples blinded to the observer, and invaded vs non-invaded parasites were quantified by eye on a DeltaVision Elite deconvolution microscope (GE Healthcare) fitted with a 100× UPlanSApo oil immersion objective lens (NA 1.40). Parasites that were both red (Tomato$^+$) and green (SAG1$^+$) were considered to be extracellular, while those that were red but not green were considered as having invaded a host cell. At least 100 parasites were counted per condition and replicate.

### *T. gondii* intracellular proliferation assay

TATiΔ*ku80*/Tomato$^+$ or *Tg*CDPK1$^{G128M}$/Tomato$^+$ strain *T. gondii* parasites were prepared in a similar way to the invasion assay. Following mechanical egress in IC buffer, parasites were counted and diluted to $5 \times 10^4$ parasites/mL in IC buffer, added to wells of a 24-well plate containing confluent HFF cells cultured on coverslips, and incubated at 37˚C for 45 min to allow the parasites to attach to host cells. IC buffer was removed and replaced with 1 mL DMEM, and parasites were allowed to invade and begin to proliferate for 4 h at 37˚C in the absence of

drug. Medium was then removed, cells were washed twice to remove uninvaded parasites, and replaced with 1 mL DMEM with either DMSO (vehicle control), MMV688853 (5 μM), 3-MB-PP1 (5 μM) or atovaquone (1 μM). Parasites were cultured for a further 19 h at 37˚C, then fixed in 3% (w/v) PFA in PBS for 15 min. Coverslips were mounted onto slides, and the identity of each was blinded to the observer. The number of parasites per vacuole were quantified on a DeltaVision Elite deconvolution microscope (GE Healthcare) by eye. At least 100 vacuoles were counted per condition and replicate. Parasite vacuoles containing parasites that deviated from the typical parasite shape were classified as 'abnormal' (S6 Fig).

## Data analysis

We performed all statistical analyses in GraphPad Prism v9. We performed paired, two-tailed t-tests to compare the difference in proliferation $EC_{50}$ values of each compound for WT vs $ATV^R$ strains in both *T. gondii* and *P. falciparum*, and for $ELQ^S$ vs $ELQ^R$ strains in *T. gondii* (Tables 1–2; Figs 7–9). We performed ANOVA followed by Tukey's multiple comparisons test to compare the inhibition of invasion (Fig 4B) or OCR (Fig 4E) by the compounds against WT and $Tg$CDPK1$^{G128M}$ *T. gondii* parasites relative to each other and to a no-drug control. We performed paired, two-tailed t-tests to compare the $EC_{50}^{OCR}$ values (Fig 4D) or proliferation $EC_{50}$ values (Fig 4F) of MMV688853 against WT and $Tg$CDPK1$^{G128M}$ *T. gondii* parasites, and to compare $EC_{50}^{OCR}$ values of inhibitors in WT vs $ATV^R$ and $ELQ^S$ vs $ELQ^R$ *T. gondii* parasites (Fig 10). We performed ANOVA followed by Dunnett's multiple comparisons test to compare the ability of compounds to inhibit *T. gondii* Complex III enzymatic activity relative to a no-drug control (Fig 5I), the ability of compounds to inhibit OCR and ECAR of WT *T. gondii* parasites relative to a no-drug control (S4 Fig), and the ability of compounds to inhibit malate-dependent, glycerol 3-phosphate-dependent and TMPD-dependent OCR relative to a no-drug control (S7 Fig).

## Supporting information

**S1 Table. Data table of the Pathogen Box screen for inhibitors of mitochondrial $O_2$ consumption in *T. gondii* parasites. (Tab 1A-B)** Plate configurations of the two Seahorse XFe96 plates used in the screen. Included are the MMV compounds that were injected into the indicated wells from Ports A-C during the first (Tab 1A) and second (Tab 1B) assays. Hit compounds are highlighted in yellow, compounds excluded from subsequent analyses because they were injected after hit compounds are highlighted in blue, compounds rescreened on the second plate are highlighted in green, background wells are highlighted in orange, and wells from a separate experiment that was included on Plate 2 are indicated in gray. Atovaquone and antimycin A was injected from Port D into all wells. **(Tab 2A-B)** Calculated OCR and ECAR values obtained for each well at each measurement time in Plate 1 (Tab 2A) and Plate 2 (Tab 2B). The time from the commencement of the assay at which each measurement was taken is indicated. Injection of compounds from Port A occurred between the third and fourth measurements, injection from Port B occurred between the seventh and eighth measurements, injection from Port C occurred between the 11th and 12th measurements, and injection from Port D occurred between the 15th and 16th measurements. Also indicated are the average OCR and ECAR values across all wells at each measurement point, with the average OCR value in the final measurement following atovaquone/antimycin A injection used to determine the non-mitochondrial OCR for the assay (highlighted in yellow). **(Tab 3)** Summary of the assay data, expressed in a table modified from the Pathogen Box plate mapping spreadsheet provided by MMV. Included are: the Pathogen Box plate and position of the test compounds; the MMV compound identification number and common name (where applicable); the Seahorse

XFe96 plate well position and injection port from which the compound was injected during the assay (with compounds injected into Plate 1 indicated in green and compounds injected into Plate 2 in yellow); the mitochondrial OCR (mOCR) values immediately before and after compound injection (calculated by subtracting the non-mitochondrial OCR from the OCR values listed in Tab 2A-B; note that the post-drug values were obtained on the fourth measurement, approximately 18 min, after compound injection for all compounds except auranofin, for which we noticed a more gradual OCR decrease, and which was therefore calculated after ~40 min); the percent inhibition of mitochondrial OCR following compound injection (calculated from the difference between pre-compound injection and post-compound injection mOCR values); an indication of whether compounds met the >30% inhibition cut-off for hit compounds; and the chemical structures of all MMV compounds expressed in simplified molecular input line entry system (SMILES) notation. **(Tab 4A-B)** The $O_2$ and pH values obtained during each measurement in the Seahorse XFe96 assays in plate 1 (Tab 4A) and plate 2 (Tab 4B). These values form the basis from which the OCR and ECAR values depicted in Tab 2A-B were determined. **(Tab 5A-B)** Calibration data for the OCR (left) and ECAR (right) assays conducted in plate 1 (Tab 5A) or plate 2 (Tab 5B). Note that the pH calibrations failed for some wells in plate 1 and we therefore did not include ECAR data in our analysis. (XLSX)

**S1 Fig. Candidate ETC inhibitors inhibit proliferation of *T. gondii* parasites. (A)** Proliferation of tdTomato-expressing *T. gondii* parasites cultured in the absence of drug (pink circles), or in the presence of atovaquone (two fold serial dilution from highest concentration (100 nM; dark green) to lowest concentration (0.39 nM; light green)) over a 6-day period. Values are expressed as a percent of the average fluorescence from the no-drug control on the final day of the experiment, and represent the mean ± SD of three technical replicates. Data are from one experiment and are representative of three independent experiments. Similar proliferation curves were obtained for each test compound. **(B-I).** Dose-response curves depicting the percent of *T. gondii* parasite proliferation in the presence of a range of concentrations of **(B)** atovaquone, **(C)** buparvaquone, **(D)** MMV671636, **(E)** auranofin, **(F)** trifloxystrobin, **(G)** azoxystrobin, **(H)** MMV024397, or **(I)** MMV688853. Values are expressed as a percent of the average fluorescence from the no-drug control at mid-log phase growth, and represent the mean ± SEM of three independent experiments, each conducted in triplicate; error bars that are not visible are smaller than the symbol. (TIF)

**S2 Fig. Candidate parasite ETC inhibitors do not impair the proliferation of human cells, with the exception of auranofin. (A-G).** Dose-response curves depicting the proliferation of human foreskin fibroblast (HFF) cells in the presence of a range of concentrations of **(A)** atovaquone, **(B)** buparvaquone, **(C)** trifloxystrobin, **(D)** azoxystrobin, **(E)** MMV024397, **(F)** MMV688853, or **(G)** auranofin. Values are expressed as a percent of the average confluence of the no-drug control at the end point of the assay (after 4 days of proliferation), and represent the mean ± SEM of three independent experiments, each conducted in triplicate; error bars that are not visible are smaller than the symbol. **(H)** Representative images of HFF cells at the end point of the assay when grown in the absence of drug (left; no drug) or in the presence of 10,000 nM auranofin (right). (TIF)

**S3 Fig. Identified compounds inhibit $O_2$ consumption in *T. gondii*. (A-B)** Traces depicting the changes in $O_2$ consumption rate (OCR) over time of intact *T. gondii* parasites incubated with no drug (pink) or with **(A)** atovaquone (ATV; two fold serial dilution from highest

concentration—10 μM, colored dark green—to lowest concentration—0.01 μM, colored light green) or (B) auranofin (AUR; two fold serial dilution from highest concentration—80 μM, colored dark green—to lowest concentration—0.08 μM, colored light green)). FCCP (1 μM) was injected into the well to uncouple electron transport from the proton gradient and thus elicit the maximal OCR. A range of concentrations of the test compounds were then injected and the inhibition of OCR measured over time. A final injection of an inhibitory concentration of atovaquone (ATVi; 5 μM) maximally inhibited mitochondrial OCR. Values represent the mean ± SD of two technical replicates from a single experiment and are representative of three independent experiments. Similar OCR inhibition traces were obtained for each test compound. (C-I) Dose-response curves depicting the percent of *T. gondii* OCR in the presence of increasing concentrations of (C) atovaquone, (D) buparvaquone, (E) auranofin, (F) trifloxystrobin, (G) azoxystrobin, (H) MMV024397 or (I) MMV688853. Values represent the percent OCR relative to the no-drug (100% OCR) and inhibitory atovaquone-treated (0% OCR) controls, and depict the mean ± SEM of three independent experiments, each conducted in duplicate; error bars that are not visible are smaller than the symbol. (J) Comparison of the $EC_{50}$ values determined for *T. gondii* OCR (OCR $EC_{50}$; this figure and Table 3) and WT *T. gondii* proliferation (proliferation $EC_{50}$; S1 Fig and Table 1). Coloring of compounds is as in Figs 1 and 3 (atovaquone, black; buparvaquone, burgundy; auranofin, orange; trifloxystrobin, pink; azoxystrobin, light blue; MMV024397, red; and MMV688853, dark blue).
(TIF)

**S4 Fig. All tested compounds except auranofin inhibit $O_2$ consumption rate but not extracellular acidification rate.** (A) $O_2$ consumption rate (OCR) and (B) extracellular acidification rate (ECAR) of *T. gondii* parasites treated with either no drug (white), atovaquone (gray; 10 μM), trifloxystrobin (pink; 10 μM), azoxystrobin (light blue; 80 μM), MMV024397 (red; 20 μM), buparvaquone (burgundy; 20 μM), auranofin (orange; 80 μM) or MMV688853 (dark blue; 20 μM), assessed using a Seahorse XFe96 flux analyzer. Bars represent the mean ± SEM of three independent experiments each conducted in duplicate. ANOVA followed by Dunnett's multiple comparisons test were performed and *p*-values are shown. A graphical output of these data comparing OCR to ECAR is depicted in Fig 3A.
(TIF)

**S5 Fig. Flow cytometry gating strategy for propidium iodide-based parasite viability assays.** (A-C) Extracellular parasites treated with (A) DMSO, (B) atovaquone, or (C) auranofin were gated away from debris using SSC-A and FSC-A parameters (left plots). Single parasites were subsequently gated using FSC-H and FSC-A parameters (centre plots). Propidium Iodide (PI) fluorescence (right plots) was used to separate viable parasites (PI negative) from non-viable parasites (PI positive). Graphs depict data from the DMSO, 10 μM atovaquone and 20 μM auranofin treatments at the 80 minute time point from a single experiment, and are representative of the gating strategy used in three independent experiments at the compound concentrations and treatment times listed in Fig 3B.
(TIF)

**S6 Fig. Parasites treated with 3-MB-PP1 and, to a lesser extent, MMV688853 exhibit abnormal cellular morphology.** Representative images of (A) WT or (B) *Tg*CDPK1$^{G128M}$ parasites expressing tdTomato observed during intracellular proliferation assays, with tdTomato fluorescence (top) and differential interference contrast (DIC; bottom) images depicted. WT or *Tg*CDPK1$^{G128M}$ parasites were cultured in the absence of drug (no drug), or the presence of MMV688853 (5 μM), 3-MB-PP1 (5 μM) or atovaquone (1 μM) for 20 h. Abnormal morphology was defined as vacuoles that contained misshapen parasites as depicted in the images of

MMV688853 and 3-MB-PP1 treated parasites. Scale bars are 2 μm.
(TIF)

**S7 Fig. Quantification of the ETC target determination assay.** Quantification of the change in the (**A**) malate- or (**B**) glycerol 3-phosphate-dependent $O_2$ consumption rate (OCR) of plasma membrane-permeabilized *T. gondii* parasites after injection of inhibitors (atovaquone, 1.25 μM; buparvaquone, 5 μM; trifloxystrobin, 2.5 μM; azoxystrobin, 80 μM; MMV024397, 20 μM; MMV688853, 20 μM; auranofin, 10 μM). (**C**) Quantification of the rescue of OCR by TMPD after inhibition by the above compounds. Data were normalized relative to the baseline OCR level pre-substrate injection (0% OCR) and the malate/G3P-elicited OCR level pre-drug injection (100%). Data represent the mean ± SEM of three independent experiments each conducted in at least duplicate. Note that the TMPD graph combines data from both the malate and G3P experiments. Error bars that are not visible are smaller than the symbol. ANOVA followed by Dunnett's multiple comparisons test were performed and *p*-values are shown. Representative traces from which these data were quantified are depicted in Fig 5.
(TIF)

**S8 Fig. Characterizing Complex III inhibition by the candidate ETC inhibitors in *T. gondii*.** (**A-G**) Complex III activity assays showing the change in absorbance of equine heart CytC at 550 nm over time (measured every 15 s) in the presence (+ para, dark shade) or absence (- para, light shade) of parasite extracts, and in the absence of drug (no drug, DMSO vehicle control, black or gray), or in the presence of (**A**) auranofin (orange, 10 μM), (**B**) atovaquone (green, 1.25 μM), (**C**) buparvaquone (burgundy, 5 μM), (**D**) trifloxystrobin (pink, 2.5 μM), (**E**) azoxystrobin (light blue, 80 μM), (**F**) MMV024397 (red, 20 μM) or (**G**) MMV688853 (dark blue, 20 μM). Decylubiquinol ($DUBH_2$) was added at the indicated times. Data are from a single experiment and are representative of three independent experiments. (**H**) Complex III activity assays showing the change in absorbance of equine heart CytC at 550 nm over time where change in absorbance in the absence of parasite extracts (*i.e.* background absorbance) was subtracted from the change in absorbance in the presence of parasite extracts. Data are from a single experiment and are representative of three independent experiments.
(TIF)

**S9 Fig. Most identified compounds inhibit $O_2$ consumption in *P. falciparum*.** (**A-F**) Dose-response curves depicting the OCR of permeabilized *P. falciparum* parasites in the presence of increasing concentrations of (**A**) atovaquone, (**B**) auranofin, (**C**) trifloxystrobin, (**D**) azoxystrobin, (**E**) MMV024397 or (**F**) MMV688853. Values represent the percent OCR relative to the no-drug (100% OCR) and atovaquone-treated (0% OCR) controls, and are depicted as the mean ± SEM of three independent experiments; error bars that are not visible are smaller than the symbol.
(TIF)

## Acknowledgments

We would like to thank the Medicines for Malaria Venture for supplying the 'Pathogen Box' compounds, Wes Van Voorhis (University of Washington) for supplying extra MMV688853 (BKI-1517) compound, John Boothroyd and Michael Panas (Stanford University) for supplying atovaquone-resistant *T. gondii* parasites, Stone Doggett (Oregon Science and Health University) for supplying ELQ-300-resistant *T. gondii* parasites, Emily Crisafulli and Stuart Ralph (University of Melbourne) for supplying yDHODH-expressing *P. falciparum* parasites, Harpreet Vohra and Michael Devoy (ANU) for assistance with flow cytometry, Michael Devoy for

assistance with establishing the XFe96 assays, Teresa Neeman from the ANU Statistical Consulting Unit for assistance with data analysis, the 2020 ANU Parasitology Course (BIOL3142) for contributing to trial experiments with MMV688853, Adele Lehane (ANU) and the ANU parasitology journal club for comments on the manuscript, and the Canberra Branch of the Australian Red Cross Lifeblood for the provision of erythrocytes and human serum.

## Author Contributions

**Conceptualization:** Jenni A. Hayward, F. Victor Makota, Rachel A. Leonard, Esther Rajendran, Christina Spry, Kevin J. Saliba, Alexander G. Maier, Giel G. van Dooren.

**Data curation:** Jenni A. Hayward, F. Victor Makota, Daniela Cihalova, Rachel A. Leonard.

**Formal analysis:** Jenni A. Hayward, F. Victor Makota, Daniela Cihalova, Rachel A. Leonard, Esther Rajendran, Soraya M. Zwahlen, Laura Shuttleworth, Christina Spry.

**Funding acquisition:** Daniela Cihalova, Esther Rajendran, Kevin J. Saliba, Alexander G. Maier, Giel G. van Dooren.

**Investigation:** Jenni A. Hayward, F. Victor Makota, Daniela Cihalova, Rachel A. Leonard, Esther Rajendran, Soraya M. Zwahlen, Laura Shuttleworth, Ursula Wiedemann, Christina Spry.

**Methodology:** Jenni A. Hayward, F. Victor Makota, Daniela Cihalova, Rachel A. Leonard, Esther Rajendran, Soraya M. Zwahlen.

**Project administration:** Christina Spry, Kevin J. Saliba, Alexander G. Maier, Giel G. van Dooren.

**Resources:** Christina Spry, Kevin J. Saliba, Alexander G. Maier, Giel G. van Dooren.

**Supervision:** Christina Spry, Kevin J. Saliba, Alexander G. Maier, Giel G. van Dooren.

**Validation:** Jenni A. Hayward, F. Victor Makota, Daniela Cihalova, Rachel A. Leonard, Soraya M. Zwahlen, Laura Shuttleworth, Ursula Wiedemann.

**Visualization:** Jenni A. Hayward, F. Victor Makota, Daniela Cihalova, Rachel A. Leonard, Esther Rajendran, Soraya M. Zwahlen, Laura Shuttleworth, Christina Spry.

**Writing – original draft:** Jenni A. Hayward, Giel G. van Dooren.

**Writing – review & editing:** Jenni A. Hayward, F. Victor Makota, Daniela Cihalova, Esther Rajendran, Soraya M. Zwahlen, Laura Shuttleworth, Christina Spry, Kevin J. Saliba, Alexander G. Maier, Giel G. van Dooren.

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
