## [Decision Letter · Decision Letter 0]

19 Jun 2022

Dear Dr. van Dooren,

Thank you very much for submitting your manuscript "A screen of drug-like molecules identifies chemically diverse electron transport chain inhibitors in apicomplexan parasites" for consideration at PLOS Pathogens. As with all papers reviewed by the journal, your manuscript was reviewed by members of the editorial board and by several independent reviewers. In light of the reviews (below this email), we would like to invite the resubmission of a significantly-revised version that takes into account the reviewers' comments.

The reviewers raised a number of issues that will need to be addressed. Importantly,  please i) provide evidence for the robustness of the assay including correlation between growth inhibition and OCR inhibition and its scalability for HTP screening, ii) consolidate the work on MMV688853 ideally with the identification of the ETC target. However while SAR based on this molecule could help to characterize the basis of its dual activity, such investigation is clearly beyond the scope of this study.

We cannot make any decision about publication until we have seen the revised manuscript and your response to the reviewers' comments. Your revised manuscript is also likely to be sent to reviewers for further evaluation.

Sincerely,

Dominique Soldati-Favre

Section Editor

PLOS Pathogens

Dominique Soldati-Favre

Section Editor

PLOS Pathogens

Kasturi Haldar

Editor-in-Chief

PLOS Pathogens

orcid.org/0000-0001-5065-158X

Michael Malim

Editor-in-Chief

PLOS Pathogens

orcid.org/0000-0002-7699-2064

Reviewer's Responses to Questions

**Part I - Summary**

Reviewer #1: In this study, Hayward et al. screened the Medicines for Malaria Venture small molecule library for activity against the T. gondii and P. falciparum electron transport chain (ETC). The Seahorse XFe96 flux analyzer successfully distinguished between the on-target inhibition of the parasite ETC by measuring oxygen consumption rates while simultaneously assessing extracellular acidification rates for off-target inhibition. The authors validate seven structurally diverse compounds (two of which are novel ETC inhibitors) from their primary screen with complementary assays. Four candidates acted on the T. gondii and three on the P. falciparum ETC Complex III, the known target for the antimalarial drug atovaquone. The dual target of MMV688853 for the ETC and TgCDPK1 is nicely demonstrated. Finally, the authors find that several compounds inhibit atovaquone-resistant T. gondii and P. falciparum parasites, suggesting a unique mode of action against the Complex III to atovaquone. This study successfully demonstrated a scalable screen of the parasite ETC, a promising avenue for drug development given the structural differences between human and parasite ETC proteins. The manuscript is nicely written and significantly contributes to the chemical biology and anti-parasitic drug development fields. While important, the work would have a greater impact to the Plos Pathogens community if the findings advanced our understanding of a molecular pathway in the pathogen and/or how the pathogen interacts with its host organism.

Reviewer #2: This work reports a screen and further characterization of small molecules from the MMV “Pathogen Box” with a focus on molecules that inhibit the electron transport chain in T. gondii and Plasmodium falciparum. The primary justification for this work is based on the fact that the ETC is a validated target in both parasites. The authors argue that their methods could be scaled to large scale screens to identify additional ETC inhibitors.

While the authors are correct, the ETC is a validated target – the two major targets previously identified as DHODH and Cytochrome b. It is well known and well-documented that many small molecules identified in P. falciparum growth assay screens target the ETC. In fact most groups do a counter-screen to identify ETC inhibitors, so the findings reported here are not surprising and are not novel. Others have screened the pathogen box and demonstrated several ETC inhibitors.

The work with the Seahorse technology is of interest and is valuable for the biochemical characterization of specific inhibitors. This paper demonstrates the Seahorse technology works with these parasites, however, this was an expected result. Noteworthy would have been the opposite result. The one set of novel experiments is the work with MMV688853. The Seahorse results provided validation that the compound inhibited ETC independent of its inhibition of TgCDPK1.

The authors go on to characterize the inhibitor MMV688853. This is by far the most interesting result in the paper and it is novel. This compound has previously been identified as a TgCDPK1 inhibitor and the work reported here strongly supports a dual action of this molecule as an ETC inhibitor in T. gondii. Interestingly MMV688853 has a different activity profile in P. falciparum. This points to differences in these systems – raising several biological questions. It also points to the potential for a dual inhibitor at least for T. gondii and SAR based on this molecule could help to characterize the basis of this dual activity.

The work demonstrating activity of selected small molecues on atovaquone resistant parasites is interesting, but unfortunately not comprehensive. Resistance mutations in the cytochrome b gene have been demonstrated at several independent sites. Collateral sensitivity has also been observed with several small molecule pairs that target cytochrome b.

The focus and the title/abstract of the manuscript are misleading – this is at best a “mini” screen and on a set of compounds that has been previously screened by many. There is no evidence for the scalability and the idea of targeting the ETC is one well visited in the literature and by many previous screening campaigns. A much stronger paper could be based on the novel findings with MMV688853. It would be very useful to identify the ETC target of this compound and further understand the difference in the inhibition in P. falciparum.

Reviewer #3: This manuscript by Hayward et al. presents an approach to screen for electron transport chain inhibitors using a Seahorse XFe96 flux analyzer. Using this approach, the authors screened the MMV pathogen box against Toxoplasma gondii Oxygen consumption rate. They further validated the hits from the screen against T. gondii proliferation, Oxygen consumption rate (OCR), ETC complex III activity, and against P. falciparum proliferation and OCR. They also further characterized the hits against parasite lines that are resistant to atovaquone.

Overall, the work in characterizing the hits as ETC inhibitors appears complete and thorough. This work provides supporting data that the electron transport chain is a valid drug target for Toxoplasma gondii and Plasmodium falciparum. The approach for finding inhibitors of ATV resistant parasites is interesting.

**Part II – Major Issues: Key Experiments Required for Acceptance**

Reviewer #1: -Given that the parasite ETC Complex III has structural differences from the human homolog, it is likely that the inhibitors discovered in this study are selective for T. gondii and P. falciparum proteins. However, the authors could provide cell viability data to demonstrate this and/or reference the selection criteria for the MMV library if this was done previously. Would it be possible to speculate on molecular mechanisms that account for this selectivity?

-For the intracellular proliferation assays shown in Fig 4, “vacuoles containing 1-8+ (gray tones) or abnormal (orange) parasites” were quantified. It is stated that abnormal morphology was defined as vacuoles that contained misshapen parasites, but how this assessment was made is not clarified in the legend or methods. Was this done by eye or based on ellipticity? Could metrics be given that defined “abnormal” and representative images be included in the SI.

Reviewer #2: The focus and the title/abstract of the manuscript are misleading – this is at best a “mini” screen and on a set of compounds that has been previously screened by many. There is no evidence for the scalability and the idea of targeting the ETC is one well visited in the literature and by many previous screening campaigns. I would redo the paper with a focus on the dual action compound discovered.

I would suggest putting much of the data reported in the manuscript in supplemental materials. I think it is valuable to show validation of the Seahorse assay, but I would do it in the context of characterizing MMV688853 mechanism of action. A much stronger paper could be based on the novel findings with MMV688853. It would be very useful to identify the ETC target of this compound and further understand the difference in the inhibition in P. falciparum. This will require additional experiments and a more detailed analysis of this compound mechanism of action for P. falciparum - similar to that which was already presented for T. gondii. SAR around MMV688853 would greatly strengthen the paper and provide insight into the dual inhibition.

Reviewer #3: Major issues:

The main concern of the protocol using the Seahorse XFe96 assay is that only 2 plates were run for the assay of 400 compounds and 3 compounds were tested sequentially in the same well without washing or removing the previous drug. Therefore, it seems that there could be potential for false positive hits. This could present problems when scaling up to do HTP screening if that is the future direction of this screen. It would be important to report Z’ score for the plates tested to show the robustness of the assay. This is relevant considering the need to use Cell Tak which most likely changes the physiology of the cells.

• Fig 5: The presentation of the data could be clearer. As presented, it is not clear the significance of the inhibition by each drug so bar graphs showing the difference between the OCR before and after adding drugs for each one should be included. Statistics should also be shown. Also, in the figure the dotted lines appear to be shifted as the change in slope occurs prior to the addition of the reagent. Fig. 6 has similar issues and also the shift of the dotted lines.

• Is there a reason for not using succinate as substrate? Previous work with Toxoplasma oxygen consumption showed that it is a better substrate than malate and much better than G3P (see table 1 of JBC273: 31040). I think that succinate is a more direct substrate. I suggest to test it at least for the experiments with Toxoplasma

• It is not clear why the authors use such high concentrations of inhibitors for the assays with permeabilized parasites when growth or oxygen consumption rates of intact parasites are inhibited at 50% at much lower concentrations. For some of the compounds like Azoxystrobin which has a growth inhibitory concentration of 310 nM the author uses 80 µM for the in vitro assay. These results may be indicating that other enzymes are being targeted in the parasites and the drugs are not specific.

• One way to show that the drugs are specifically inhibiting the ETC is to show a correlation between growth inhibition to OCR inhibition. It will become clear if there is or there is no correlation. From the presented data it is hard to predict but for example buparvaquone inhibits growth with a 0.7 nM EC50 and the OCR 50% inhibition is 1.18 micromolar. Atovaquone needs 10 nM for inhibiting growth at 50% while only needs 180 nM to inhibit OCR. It appears as if buparvaquone would be inhibiting other targets in the parasites. More confusion is added using buparvaquone at 5 micromolar for the assays with permeabilized parasites and the assay of complex III enzymatic activity. The case of Azoxystrobin is also puzzling as it inhibits 50% growth at 310 nM, OCR at 7 micromolar and for the in vitro assays the authors use 80 micromolar. I suggest to test lower concentrations of these compounds for the experiments with permeabilized cells.

• The inhibition of Plasmodium OCR is stronger for most compounds, but authors use 10 micromolar for all the in vitro assays.

• Auranofin which the authors claim may not be targeting the mitochondria ETC shows a 50% inhibition of the OCR at 2.48 micromolar which is similar to the MMV024397 which the authors claim does inhibit the ETC. This needs to be addressed in the discussion.

**Part III – Minor Issues: Editorial and Data Presentation Modifications**

Reviewer #1: -It would be helpful to the readers to introduce the Seahorse XFe96 flux analyzer as this is not a common instrument.

-Given the complex lifecycles of T. gondii and P. falciparum, the authors should refer to specific cell types and stages of the parasite lifecycles rather than saying ‘host cells’ for in vitro experiments.

-While the authors determine that several of their compounds do not cross-react with atovaquone resistant parasites with mutations at the cytochrome b protein, this does not confirm that the parasites would not develop drug resistance with a different mutation. While it is mentioned in the methods that the authors selected for clonal populations for a single atovaquone mutation, it should be made clear in the main text that this study was performed with only one of the known mutation sites in cytochrome b. Performing assays with additional cytochrome b mutations as cited in the discussions or binding studies would provide more confidence that these compounds have novel modes of action outside of the atovaquone binding site.

Reviewer #2: (No Response)

Reviewer #3: • Table 1 & 2 (and throughout): IC50 is usually used for inhibition of a specific target like inhibition of an isolated enzyme. For growth assays EC50 is more appropriate. For the inhibition of oxygen consumption, it would be appropriate to use O2-EC50. It is not clear if Table 2 is showing the inhibition of OCR in intact parasites or permeabilized parasites. The protocol used is not clarified in the Table legend.

• The method section is missing the statistical analyses of the data and also clarification of the number of experiments and replicates for all assays.

• For the flow cytometry methods, it says that data are exported for further analysis using FlowJo but there is no description of what analysis was done.

• Fig 2: Statistical analysis of WT vs yDHODH parasites is missing

• Fig 3: Could use a 2-way Anova and compare the drug treated vs the control.

• Fig 4C: The colors of the no drug bars are difficult to differentiate between 4, 8, and 8+ parasites per vacuole.

PLOS authors have the option to publish the peer review history of their article (what does this mean?). If published, this will include your full peer review and any attached files.

Reviewer #1: No

Reviewer #2: No

Reviewer #3: No
---

## [Decision Letter · Decision Letter 1]

27 May 2023

Dear Dr. van Dooren,

Thank you very much for submitting your manuscript "A screen of drug-like molecules identifies chemically diverse electron transport chain inhibitors in apicomplexan parasites" for consideration at PLOS Pathogens. As with all papers reviewed by the journal, your manuscript was reviewed by members of the editorial board and by several independent reviewers. The reviewers appreciated the attention to an important topic. Based on the reviews, we are likely to accept this manuscript for publication, providing that you modify the manuscript according to the review recommendations.

The remaining issues raised by Reviewer #3 can in my view be addressed in a minor revision by changes in the text, and depositing all the raw data .

- The risk of interference caused by addition of the second and third drug should be stated as a potential weakness. While this could lead to 'overlooking' potentially interesting candidates, it does not change the fact that the 7 candidates identified were individually tested and thoroughly investigated for their target and MoA.

- The limitation in regard of scalability of the screen is already addressed. It identifies compounds with a certain MoA and should not be compared to screens that 'just' identify an EC50 (measurement of replication/fitness).

- ECAR value is shown as a reference in the Figure S4. The authors should deposit the data from all screens on a data depository.

Sincerely,

Dominique Soldati-Favre

Section Editor

PLOS Pathogens

Dominique Soldati-Favre

Section Editor

PLOS Pathogens

Kasturi Haldar

Editor-in-Chief

PLOS Pathogens

orcid.org/0000-0001-5065-158X

Michael Malim

Editor-in-Chief

PLOS Pathogens

orcid.org/0000-0002-7699-2064

Reviewer Comments (if any, and for reference):

Reviewer's Responses to Questions

**Part I - Summary**

Reviewer #1: In this revised manuscript Hayward et al have addressed many of the reviewer suggestions. The manuscript describes a new screen to identify apicomplexan ETC inhibitors and provides chemical diverse scaffolds from the Pathogen Box that target the parasite ETC.

Reviewer #3: This manuscript by Hayward et al. presents an approach to screen for electron transport chain inhibitors using a Seahorse XFe96 flux analyzer. Using this approach, the authors screened the MMV pathogen box against Toxoplasma gondii oxygen consumption rate. They further validated the hits from the screen against T. gondii proliferation, oxygen consumption rate (OCR), ETC complex III activity, and against P. falciparum proliferation and OCR. They also further characterized the hits against parasite lines that are resistant to atovaquone and in this revision they included ELQ resistant lines.

The main strength of the work is the discovery of new ETC inhibitors that are effective against toxoplasma and plasmodium.

The main weakness is the limitation of the assay.

This is a re-submission and the authors made a great effort to address the concerns of the reviewers.

However, some issues remain.

**Part II – Major Issues: Key Experiments Required for Acceptance**

Reviewer #1: (No Response)

Reviewer #3: The assay has limitations. The Z’ score is not great and can barely distinguish a hit from a non-hit. A good Z’ factor is usually >0.5. The main reason could be the addition of a second and a third drug to the same well.

The second and the third addition are tested under different conditions. What about if there is antagonism between drugs? Or synergy? What about if a drug causes detachment of the parasites? This could also result in false negatives or result in variability. This is a problem with the assay, and the authors would need to acknowledge it.

I think that the screen is a weakness of the work and according to the data presented it may not be easily scalable.

The authors mentioned that the main advantage of the screen is the simultaneous measurement of OCR and extracellular acidification rate (ECAR) but they do not show this second part. Figure 1 only shows the OCR and they only measured ECAR for the selected hits. Is it possible to measure ECAR at the same time as OCR for the whole plate?

The rest of the work is good as the authors discovered new ETC inhibitors that are also effective against plasmodium. I would re-focus the work on the characterization of the mechanism of inhibition of these new drugs instead of focusing on the screening which is a weakness.

**Part III – Minor Issues: Editorial and Data Presentation Modifications**

Reviewer #1: (No Response)

Reviewer #3: • The EC50 for growth and for Oxygen rate could be differentiated by using O2-EC50. This distinction will add clarity to the description of the results. As presented it is not clear to which EC50 they are referring

• In the description of Figure 2 the letters are not mentioned.

• Is there a reason why the EC50 for the OCR measurements between ATOs vs ATOr lines in Table 3 was not determined? May be re-design the table so that there are not so many NDs?

• Figure S7C: a positive control with antimycin a and a negative control with a drug that inhibits downstream to Cyt c would be appropriate as the figure shows all drugs that rescue.

• Concerning the response to number 20 in the response. Using permeabilized plasmodium may give false positives for drugs that are not able to enter the parasite. It would not be comparable to the OCR measurements of T. gondii.

• The discussion needs some work as it is repetitive of the results. It even mentions the figures again.

PLOS authors have the option to publish the peer review history of their article (what does this mean?). If published, this will include your full peer review and any attached files.

Reviewer #1: No

Reviewer #3: No

Figure Files:

Data Requirements:

Reproducibility:

References:

---

## [Editor Report · Decision Letter 2]

28 Jun 2023

Dear Dr. van Dooren,

We are pleased to inform you that your manuscript 'A screen of drug-like molecules identifies chemically diverse electron transport chain inhibitors in apicomplexan parasites' has been provisionally accepted for publication in PLOS Pathogens.

Best regards,

Dominique Soldati-Favre

Section Editor

PLOS Pathogens

Dominique Soldati-Favre

Section Editor

PLOS Pathogens

Kasturi Haldar

Editor-in-Chief

PLOS Pathogens

orcid.org/0000-0001-5065-158X

Michael Malim

Editor-in-Chief

PLOS Pathogens

orcid.org/0000-0002-7699-2064
---

## [Editor Report · Acceptance letter]

17 Jul 2023

Dear Dr. van Dooren,

We are delighted to inform you that your manuscript, "A screen of drug-like molecules identifies chemically diverse electron transport chain inhibitors in apicomplexan parasites," has been formally accepted for publication in PLOS Pathogens.

Best regards,

Kasturi Haldar

Editor-in-Chief

PLOS Pathogens

orcid.org/0000-0001-5065-158X

Michael Malim

Editor-in-Chief

PLOS Pathogens

orcid.org/0000-0002-7699-2064